# Jag1-Notch *cis*-interaction determines cell fate segregation in pancreatic development

Xiaochan Xu [1], Philip Allan Seymour[2,3], Kim Sneppen[1], Ala Trusina [1], Anuska la Rosa Egeskov-Madsen [2,3], Mette Christine Jørgensen [2,3], Mogens Høgh Jensen [1] ✉ & Palle Serup [2,3] ✉

The Notch ligands Jag1 and Dll1 guide differentiation of multipotent pancreatic progenitor cells (MPCs) into unipotent pro-acinar cells (PACs) and bipotent duct/endocrine progenitors (BPs). Ligand-mediated *trans*-activation of Notch receptors induces oscillating expression of the transcription factor Hes1, while ligand-receptor *cis*-interaction indirectly represses Hes1 activation. Despite Dll1 and Jag1 both displaying *cis*- and *trans*-interactions, the two mutants have different phenotypes for reasons not fully understood. Here, we present a mathematical model that recapitulates the spatiotemporal differentiation of MPCs into PACs and BPs. The model correctly captures cell fate changes in Notch pathway knockout mice and small molecule inhibitor studies, and a requirement for oscillatory Hes1 expression to maintain the multipotent state. Crucially, the model entails cell-autonomous attenuation of Notch signaling by Jag1-mediated *cis*-inhibition in MPC differentiation. The model sheds light on the underlying mechanisms, suggesting that *cis*-interaction is crucial for exiting the multipotent state, while *trans*-interaction is required for adopting the bipotent fate.

The establishment of cell fate is typically governed by positional cues during embryonic development such that cell type specification and spatial arrangement are coupled and happen simultaneously during organ and tissue formation. Self-organization through cell-cell contact-dependent communication is repeatedly observed when multipotent progenitors break symmetry and commit to different subsequent fates[1]. Pancreas development represents an example of such a process. As the pancreatic buds emerge during embryogenesis, they quickly generate different cell types[2]. In mice, such cell fate segregation begins at E9.0 when multipotent progenitor cells (MPCs) first give rise to a few endocrine cells and subsequently, from E10.5 to E12.5, differentiate into unipotent pro-acinar cells (PACs) and bipotent progenitor (BP) cells, which later give rise to duct and endocrine cells. These transitions are governed by a gene regulatory network (GRN), where Notch signaling plays a central role[3,4]. Notch signaling is activated via intercellular *trans*-interactions between ligands and receptors, but can also

be inhibited by *cis*-interactions, where ligands sequester available receptors in an unproductive fashion[5].

Hes1 is a transcription factor directly activated by Notch signaling in the core GRN, as a consequence of ligand-mediated *trans*-activation from neighboring cells. The transcription factor Ptf1a activates the expression of both Dll1 and Jag1 in MPCs[6,7], while Hes1 inhibits ligand expression, either directly via suppression of Dll1 transcription or indirectly via suppression of Ptf1a[8,9]. For Dll1, Hes1 also directly binds to *Dll1 cis*-regulatory elements[8] and ligand degradation is facilitated by ligand-receptor *cis*-interaction[10].

After the PAC-BP cell fate decision, Ptf1a becomes highly expressed in PACs, in which Hes1 is downregulated, and these PACs thus have high expression of Dll1 and Jag1. In contrast, the BP cells express a high level of the Notch target genes Hes1, Nkx6-1, and Sox9, and low to absent levels of Ptf1a, Dll1, and Jag1. This sets the stage for a subsequent round of Neurogenin3-driven, Dll1-mediated lateral inhibition,

[1]The Niels Bohr Institute, University of Copenhagen, DK-2100 Copenhagen Ø, Denmark. [2]Novo Nordisk Foundation Center for Stem Cell Biology (DanStem), University of Copenhagen, DK-2200 Copenhagen N, Denmark. [3]Novo Nordisk Foundation Center for Stem Cell Medicine (reNEW), University of Copenhagen, DK-2200 Copenhagen N, Denmark. ✉e-mail: mhjensen@nbi.ku.dk; palle.serup@sund.ku.dk

which selects endocrine precursor cells from the BP population[11–15]. However, in this paper, we will focus only on Notch-regulated proximodistal (PD) patterning of MPCs into PACs and BPs.

PAC and BP fates are adopted with distinct spatial patterns. At E11.5, emerging PACs can be found in central parts of the epithelium but most are located on the surface of the developing organ. From E12.5 and onward, PACs largely become confined to the tips of the now branching epithelium[16]. Conversely, cells inside the epithelium, and later in the trunk domain of the branched epithelium, mostly express the BP markers, Nkx6-1 and Sox9, and seldom express Ptf1a. The two cell fates thus show a distinct PD distribution. How the developing pancreas resolves a local "salt-and-pepper"-like pattern into the global PD pattern is still an open question, but it is likely to involve differential cell migration[17].

Theoretically, the lateral inhibition mechanism may facilitate the adoption of different cell fates in a spatial "salt-and-pepper" pattern[18] in initially equivalent cells. However, Dll1-mediated lateral inhibition alone is not sufficient to bifurcate cell fates in the context of pancreatic PAC-BP development. For instance, in Jag1 deficient mutants where lateral inhibition is mediated by Dll1-Notch interactions alone, the MPCs fail to establish the BP-PAC cell fate segregation in a timely fashion[16]. First observed in *Drosophila*, the cell-autonomous *cis*-inhibition of Notch receptors by ligands prevents receptor activation when ligands are in stoichiometric excess[19]. This *cis*-inhibition can generate a switch between different cell states independent of *trans*-activation but is most often thought to reduce or prevent *trans*-activation by sequestering free receptors[5]. Based on the expression patterns of the different Notch pathway components, it was proposed that pancreatic PAC-BP segregation was dependent on Jag1-mediated cis-inhibition in emerging PACs[16], but this notion was not rigorously tested. Similarly, the underlying causes of the different effects of Dll1 and Jag1 on cell fate segregation remain unclear. These are fundamental questions yet to be resolved to understand how Notch signaling controls pancreatic development.

Downstream of Notch activation, Hes1 was recently observed to display oscillatory expression in pancreatic progenitors[16]. The oscillation period was found to be ≈90 ± 30 min in both MPCs and BPs. Dll1, being a target of Hes1, was found to oscillate with the same period and Dll1 oscillations appear to be critical for normal growth of the MPCs, while experimentally induced changes in Hes1 oscillation parameters are associated with changes in cell fate[16]. However, a direct role of Hes1 oscillations in cell fate decisions remains to be demonstrated (for a recent review of protein oscillations see ref. [20]).

In this paper, we develop a mathematical model to simulate the differentiation of MPCs into PACs and BPs by coupling the gene regulatory network to the spatial distribution of interacting cells. Our model recapitulates the spontaneous transition from a single multipotent progenitor cell type to two mutually exclusive progenitor cell types with more limited potency. It explains the spatial distribution of cell fates observed in vivo. The model predicts changes in cell type proportions consistent with experimental results obtained by genetic analyses or by chemical perturbations of Notch signaling. Remarkably, a theoretical analysis of the relative contributions of *cis*- and *trans*-interactions uncovered a critical role of Jag1-mediated *cis*-inhibition for the segregation of cell fates, highlighting the *cis*-interaction as the driving force for cell fate bifurcation in pancreatic MPC differentiation.

## Results

### Spatial expression of pancreatic cell fate deciding genes

We first examined the expression of Notch pathway components and the key transcription factors involved in PAC-BP cell fate segregation. Consistent with prior observations[16] we found that 94.3 ± 2.9% (mean ± SD, $N = 2$) of the E10.5 Ptf1a$^+$ epithelial cells co-expressed Nkx6-1 (Fig. 1a, b). Jag1 was uniformly expressed in 98.3 ± 0.5% (mean ± SD, $N = 4$) of the Ptf1a$^+$ MPCs, while Dll1 was expressed in

54.8 ± 3.4% (mean ± SD, $N = 5$) (Fig. 1a, b). 98.7 ± 1.8% (mean ± SD, $N = 2$) of the Ptf1a$^+$ MPCs were Hes1$^+$, but notably, both Hes1$^{Hi}$ and Hes1$^{Lo}$ cells could be distinguished within this population (Fig. 1a, b). That Dll1 and Hes1 are heterogeneously expressed is expected due to their oscillating expression pattern[16]. Two days later, the PD pattern of PAC and BP fates is largely established with Ptf1a$^+$ cells mainly found distally in the branching epithelium and Nkx6-1$^+$ cells located more proximally (Fig. 1c, top panel). In contrast, when Dll1 and Hes1 expression is examined with PAC and BP markers at E12.5, we find that the Dll1$^{Hi}$ and Hes1$^{Hi}$ state is mutually exclusive in both PAC and BP domains (Fig. 1c). By cell counting we found that the vast majority of emerging PACs were Ptf1a$^+$Jag1$^+$ (94.8 ± 4.0%, mean ± SD, $N = 3$) and roughly one third were Ptf1a$^+$Jag1$^+$Dll1$^-$ (29.0 ± 10.0%, mean ± SD, $N = 3$). Conversely, most BPs were Nkx6-1$^+$Jag1$^-$Dll1$^-$ (71.0 ± 4.8%, mean ± SD, $N = 3$). Instead, the nascent BPs are typically Hes1$^+$Nkx6-1$^+$ at E12.5 (79.7 ± 4.4%, mean ± SD, $N = 3$), while only 22.4 ± 9.1%, mean ± SD, $N = 3$, of the PACs are Hes1$^+$Ptf1a$^+$ (Fig. 1c–e).

Together with published Ptf1a ChIP-seq data[6,7], this ligand expression pattern suggests that Ptf1a is a direct activator of Dll1 and Jag1 expression in MPCs and PACs, but the relative importance of Ptf1a for Dll1 and Jag1 expression is unknown. Therefore, and to further test the functional requirement for Ptf1a to activate the expression of Dll1 and Jag1, we examined E12.5 heterozygous and homozygous *Ptf1a* mutants by quadruple IF staining for expression of either Jag1 or Dll1 in combination with Sox9, Cdh1, and Gcg and compared to wild-type litter mates. Remarkably, we found that epithelial Jag1 expression was reduced in heterozygotes and lost in homozygotes, while Dll1 expression was unaffected in heterozygotes and reduced in homozygotes (Fig. 1f and Supplementary Fig. 1a). This shows that Jag1 expression is strongly dependent on Ptf1a and more sensitive to reduced Ptf1a levels than Dll1 expression is.

Oscillatory expression of Dll1, and by inference, Hes1 is crucial for normal pancreas development[16,20]. To better illustrate and have an integral view of the Hes1 oscillations, we reanalyzed the expression of Hes1 in all the individual cells at different time points. We linearly scale the gene expression between the minimum and maximum levels in each cell and align all the cells according to their first peaks (Fig. 1g). With the normalized expression, both heatmaps and mean plots present two peaks of Hes1 oscillation within 200 min (Fig. 1g, h), consistent with the statistics about the period of Hes1 oscillation in Seymour et al.[16]. The second peaks shift between the cells, indicating the variability of these oscillations. The variability leads to a damped pattern for the second peaks shown in the mean plots.

Taken together, our data, the published data on Dll1 and Jag1 being direct Ptf1a target genes and Dll1 and Ptf1a being direct Hes1 target genes[6–8,16,21,22], and classical interactions within the Notch pathway suggest these variables are wired into a GRN that controls PAC-BP fate choice (Fig. 1i).

### Dll1 maintains MPC fate and Jag1 enables cell fate choice

To test how such a GRN can regulate cell fate segregation and spatial arrangement of PACs and BPs, we built a mathematical model to simulate pancreatic development in silico. Our model outlines five variables: Hes1, Dll1, Jag1, Notch, and Ptf1a, in the core GRN (Fig. 1i, see Methods). The mathematical description of the gene regulations and parameters are calibrated with empirical data from the components' expression patterns in wild-type mice, Ptf1a and Hes1 ChIP-seq and classical interactions between Notch pathway components. Additive terms are used to integrate the inputs of Hes1 and Ptf1a to Dll1 production since Dll1 is not completely abolished in the pancreas of Ptf1a knockout mice (see above). Longer degradation time of Ptf1a and Jag1 than the other variables is implemented as the two proteins have been reported having half-lives in the range of several hours[23,24], which is confirmed by our measurements in cycloheximide treated 266-6 cells (Supplementary Fig. 1b–d).

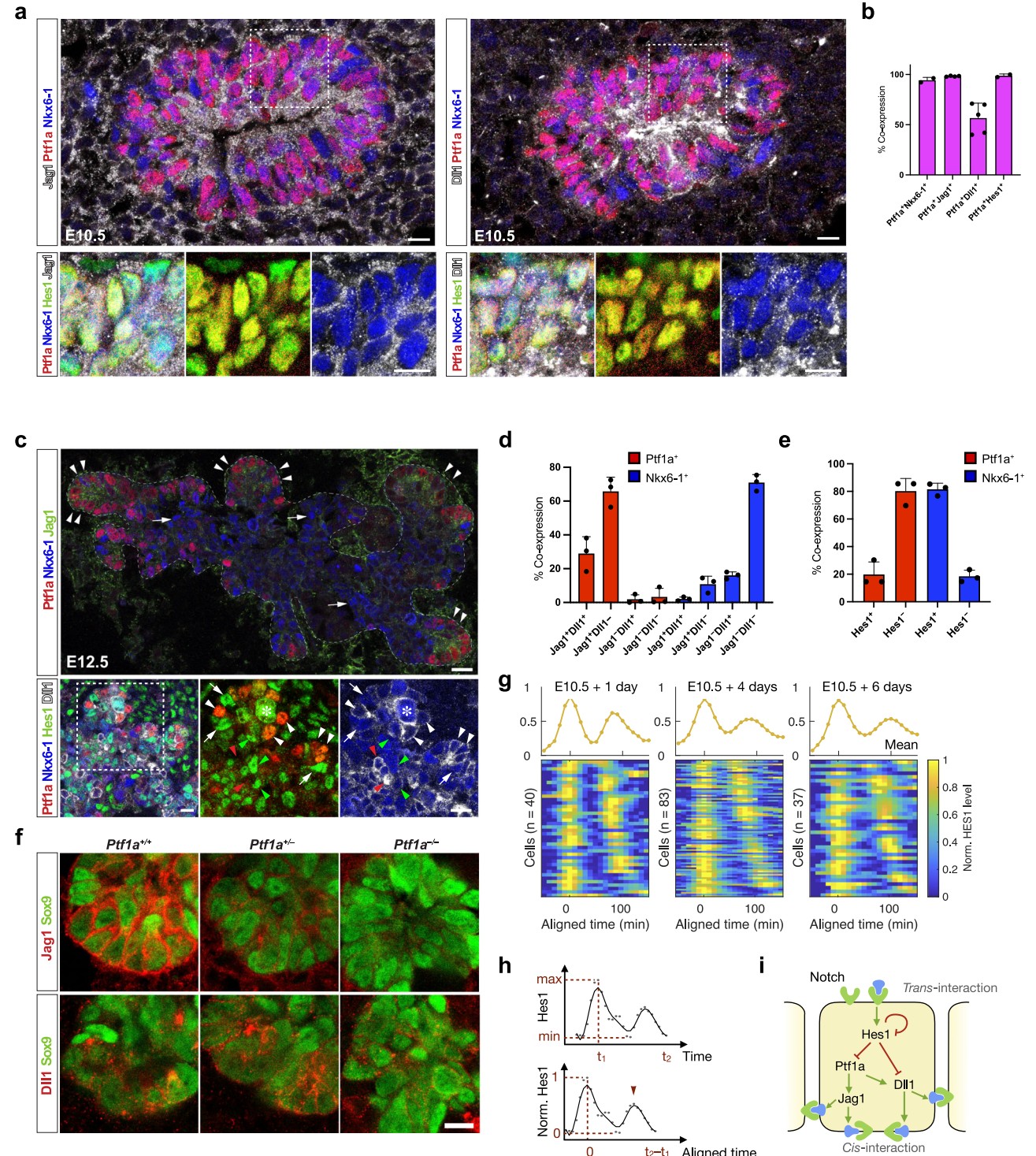

However, whether *cis*-inhibition is the mechanism by which Jag1 inhibits Notch activation in the early MPCs needs to be resolved, as this is crucial to our model. An alternative mechanism is a competition for receptors in *trans*, assuming that Jag1 is a weaker *trans*-activator than Dll1, which causes a reduction of Notch signaling by receptor sequestration[25]. We therefore generated *Foxa2*-iCre-induced conditional *Dll1; Jag1* double mutants (*Dll1; Jag1*$^{\Delta Foxa2}$) and analyzed early pancreatic bud size in these compared to controls as well as *Dll1*$^{\Delta Foxa2}$ and *Jag1*$^{\Delta Foxa2}$ mutants, initially reported in Seymour et al. 2020[16]. Crucially, the two mechanisms have different predictions for the double mutant phenotype. If Jag1 acts via *cis*-inhibition, we expect the

*Dll1; Jag1*$^{\Delta Foxa2}$ mutant to have the same phenotype as the *Dll1*$^{\Delta Foxa2}$ mutant or possibly slightly larger bud size if there is residual *trans*-activation present in the double mutant buds (e.g. from Dll4-expressing Ngn3+ cells[26,27]. Conversely, if competition for receptors in *trans* is the mechanism, then we would expect that the double mutant will be more severely reduced in size than the *Dll1* single mutant, due to the further loss of Jag1-mediated *trans*-activation in the double mutant. Importantly, we observed a slight increase in bud size in *Dll1; Jag1*$^{\Delta Foxa2}$ mutants compared to *Dll1*$^{\Delta Foxa2}$ mutants (Fig. 2a), which is consistent with a *cis*-inhibitory mechanism but argues against a mechanism involving competition for receptors in *trans*, assuming

**Fig. 1 | Expression patterns of proteins associated with cell fate segregation in pancreatic development. a** Left: Section of E10.5 dorsal pancreas stained for Jag1 (white), Hes1 (green), Ptf1a (red), and Nkx6-1 (blue). Insets show enlarged views of the boxed area. Note overlapping expression of Hes1, Jag1, Ptf1a, and Nkx6-1 in most cells. Right: Section of E10.5 dorsal pancreas stained for Dll1 (white), Hes1 (green), Ptf1a (red), and Nkx6-1 (blue). Insets show enlarged views of the boxed area. Note the heterogeneous expression of Dll1. Scale bars: 10 $\mu m$. **b** Quantification of Nkx6-1, Jag1, Dll1 and Hes1 co-expression in E10.5 Ptf1a$^+$ epithelial cells. Mean+SD. $N$ = 2, 4 and 5 embryos as indicated by the dot plots. **c** Top panel: Section of E12.5 dorsal pancreas stained for Ptf1a (red), Nkx6-1 (blue) and Jag1 (green). The dashed line encircles the epithelium. Arrowheads point at Jag1$^+$Ptf1a$^+$ tip cells and arrows at Jag1$^-$Nkx6-1$^+$ trunk cells. Scale bar: 20 $\mu m$. Bottom panels: Section of E12.5 dorsal pancreas stained for Ptf1a (red), Nkx6-1 (blue), Hes1 (green) and Dll1 (white). Note that peripheral Dll1$^{Hi}$ cells are typically Hes1$^{Lo}$Ptf1a$^+$Nkx6-1$^{Lo/-}$ (white arrowheads). Conversely, peripheral Dll1$^{Lo}$ cells are typically Hes1$^{Hi}$Ptf1a$^-$Nkx6-1$^+$ (white arrows). More centrally, Dll1$^{Hi}$ cells are typically Hes1$^{Lo}$Ptf1a$^-$Nkx6-1$^+$ (red arrowheads), while Dll1$^{Lo}$ cells are typically Hes1$^{Hi}$Ptf1a$^-$Nkx6-1$^+$, as also seen in the periphery (green arrowheads). * indicates mitotic cell. Scale bar: 10 $\mu m$. The experiment was repeated with the same result on four embryos. **d** Quantification of Dll1 and Jag1 co-expression in E12.5 Ptf1a$^+$ (red bars) and Nkx6-1$^+$ (blue bars) epithelium. Mean+SD. $N$ = 3 embryos. **e** Quantification of Hes1 co-expression in E12.5 Ptf1a$^+$ (red bars) and Nkx6-1$^+$ (blue bars) epithelium. Mean+SD. $N$ = 3 embryos. **f** Expression of Jag1 and Dll1 ligands as indicated (red) in Sox9$^+$ pancreas epithelium (green) in E12.5 wild type as well as heterozygous and homozygous *Ptf1a* mutant embryos. Scale bar: 10 $\mu m$. The experiment was repeated with the same result on two sets of embryos. **g** Oscillatory Hes1 expression during MPC differentiation. The luminescence values of the tracked cells were measured every 10 min and followed for more than 150 min. Each row of the heatmaps represents one cell's dynamic. The curves above represent the mean of the normalized values at each time point. **h** Example for normalization of Hes1 expression in one individual cell. The absolute luminescence value is linearly scaled between its minimum and maximum after the background values were subtracted. The time of each cell is aligned according to the first peak. **i** Core gene regulatory network (GRN) motif of MPC differentiation including *trans*-interaction and *cis*-interaction. The GRN includes five variables where Hes1 inhibits itself, Ptf1a and Dll1 whereas Ptf1a promotes both Jag1 and Dll1.

that additional ligands do not contribute a significant amount of *trans*-activation (see Discussion).

After substantiating the basic assumptions of our model, we next performed simulations. Remarkably, by implementing the model with just two interacting cells (Fig. 2b), we can obtain gene expression dynamics and cell fate segregation that are consistent with experimental observations, and by changing the strength of *cis*- or *trans*-interactions (Fig. 2c, d), we explored how cell fate segregation is affected by these parameters.

The auto-inhibition of Hes1 produces oscillations through a time delay motivated by delays in transcription and translation[28–31]. When the *cis*- and *trans*-interactions are at medium levels ($K_2 = 0.5$, $\gamma_1 = 0.25$), the cells go through a transient MPC state (lasting over the time interval indicated by arrows) after which they segregate into different cell fates (Fig. 2e). In these segregated states, the initially comparable amplitudes of the Hes1 oscillations differentiate into low and high amplitude states. The cell with high amplitude shows BP identity with low Ptf1a, Jag1 (Supplementary Fig. 2a, b, blue curves), and Dll1 (Fig. 2e, blue curves). The other cell with low Hes1 shows PAC identity with high Ptf1a, Jag1 (Supplementary Figs. 2a, b, red curves), and Dll1 (Fig. 2e, red curves).

Oscillation patterns of Hes1 in the two interacting cells are modulated by *trans*- and *cis*-interactions in the following way: With strong *trans*-interaction ($K_2$ is 0.15 compared with 0.5 for wild type) or weak *cis*-interactions ($\gamma_1$ is 0.1 compared with 0.25 for wild type), Hes1 exhibits comparable anti-phase oscillations in two interacting cells (Fig. 2f, g), indicating that both cells maintain the same fate. Finally, when the *trans*-interaction is weak ($K_2$ is 0.9 compared with 0.5 for wild type) or *cis*-interaction is strong ($\gamma_1$ is 0.4 compared with 0.25 for wild type), the cell fates bifurcate and one or both cells exit the MPC fate and differentiate into the downstream cell fates (Fig. 2h, i, Supplementary Fig. 2a, b).

The results shown in Fig. 2e, with intermediate *trans*- and *cis*-interactions, are consistent with the MPC fate being a transient state of the system before cell fate segregation occurs, as observed in vivo where MPCs maintain a homogeneous multipotent fate for 1-2 days only, before initiating the segregation into PAC and BP fates. With different strength of *trans*- and *cis*-interactions, we also identify different impacts of deficiency in Dll1 and, respectively Jag1 (Supplementary Fig. 2c–f). Without Dll1, the two-cell system succeeds in the cell fate segregation in spite of lower *cis*- or higher *trans*-interaction strengths compared to wild-type (compare Supplementary Fig. 2d, e), while it always fails with Jag1 deficiency (compare Supplementary Fig. 2d, f).

We next analyzed the effect of altering the transcriptional delay of Dll1 in our model. In vivo, a 6-min acceleration or delay in the appearance of Dll1 protein, achieved by modulating the transcriptional delay through altered intron-exon structure of *Dll1*, severely dampens

the oscillations of both Dll1 and Hes1 in neural progenitors and causes increased neuronal differentiation and reduced progenitor expansion[32], and these *Dll1* mutants (*type1 Dll1* and *type2 Dll1*) also show reduced expansion of pancreatic MPCs[16]. To test whether Dll1 and Hes1 oscillations in our model are also sensitive to prolonged transcriptional delay we derived a model to capture the effects of delay in *Dll1* transcription by removing Jag1 from the current model and embedding a time delay parameter ($\tau$) in the production terms of Dll1 (Supplementary Fig. 2g). The derived model shows "oscillation death" when the delay is within a short time window around 10 min. While the time delay increases, the oscillation reoccurs with a different frequency, and the two cells are in-phase rather than out of phase (Supplementary Fig. 2h). Thus, the two-cell model and its parameters are applicable both in the pancreas and neuronal system. The response of Dll1 to Hes1 levels contributes to the oscillatory gene expression in our model, and perturbation of Dll1 expression affects the coupling of the Notch signaling between interacting cells.

Overall, the two-cell system suggests that Dll1 is important for MPC maintenance while Jag1 facilitates cell fate segregation.

## A spatiotemporal model for proximodistal pancreas patterning

As the pancreas develops it is influenced by a mix of biological and physical stimuli. The early epithelium is surrounded by mesenchymal cells that support the expansion of the epithelium, with the two tissues being separated by a basement membrane. To mimic the positional cues present in the developing pancreatic anlage with their complement of epithelial and mesenchymal cells, we created a three-dimensional (3D) structural model that included cell-cell interactions (Fig. 3a) (see Methods). We assumed that the basement membrane prevents cell-cell contact-dependent signaling between mesenchyme and epithelium. This model recapitulates global spatiotemporal processes occurring during MPC differentiation by coupling the gene expression dynamics in a given cell with its local interactions with neighboring cells. During a simulation, a homogeneous group of MPCs self-organize into a structure resembling a "salt-and-pepper" pattern, but with PACs located mainly on the surface of the spherical epithelium (Fig. 3b, Supplementary Movie 1).

When simulating 60 h of development (from E10 to E12.5) with the multi-cell model, the cells initially show Dll1 and Hes1 oscillations in anti-phase, characterizing the undifferentiated MPC state. At about 10 h, the cells start to bifurcate dependent on the interactions with their neighbors. Consistent with the two-cell model, some cells develop oscillatory Hes1 expression with a high amplitude and low Dll1 and Jag1 expression, implying a BP fate, while low Hes1 and high Dll1 and Jag1 expression are observed in different cells, consistent with these adopting a PAC fate (Fig. 3c). The PACs have high Ptf1a

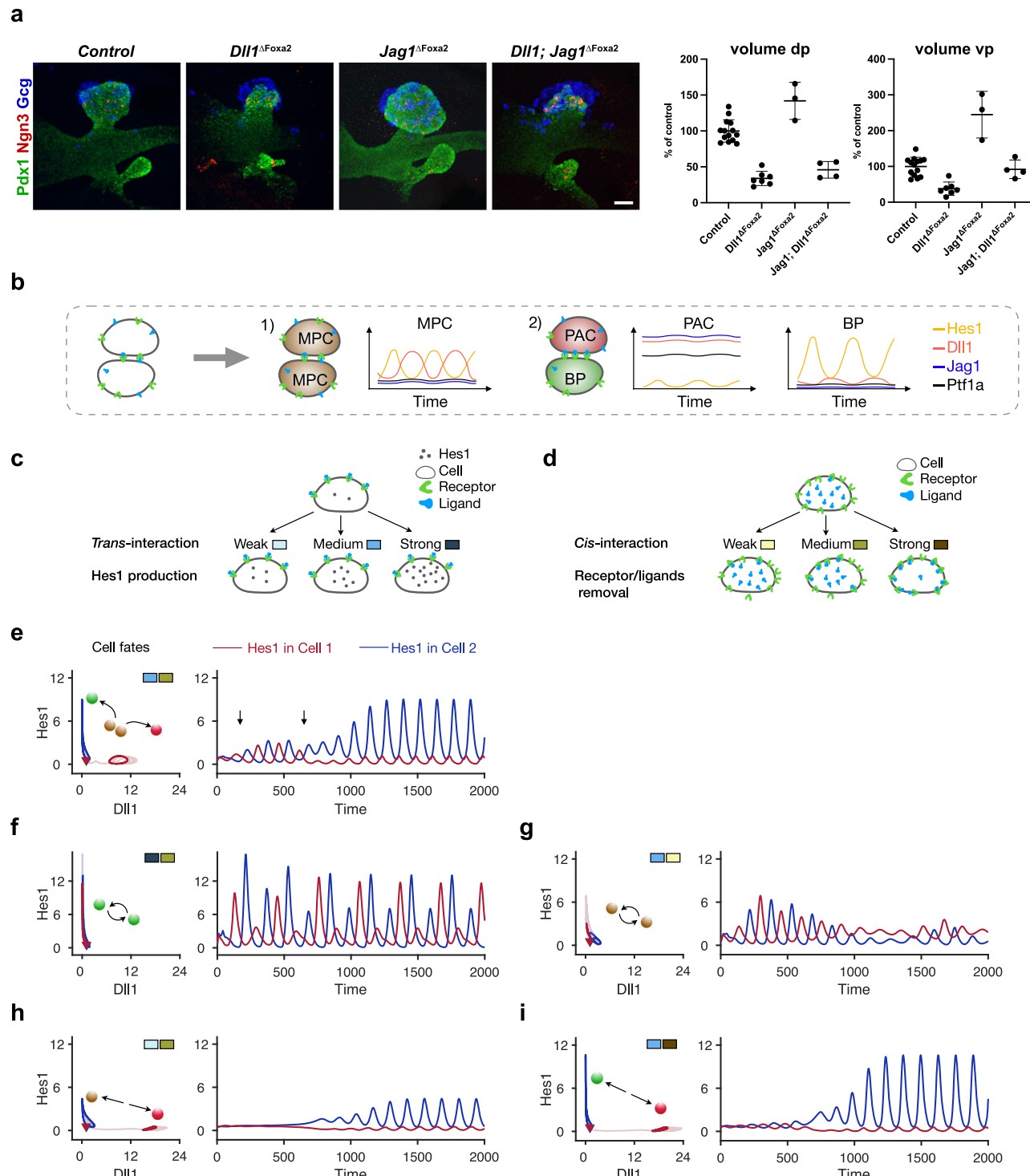

expression (the average is 9.4 and 47.2 times higher than in MPCs and BPs, respectively) as its repressor Hes1 is low (Supplementary Fig. 3a). In the multi-cell model, the anti-phase oscillations enting in the two-cell model are also observed between interacting cells (Supplementary Fig. 3b, c).

In line with experimental observations, 65% (average 67% in real embryos) of the cells adopt a BP fate, and 20% (average 22% in real embryos) of the cells are PACs at E12.5 (Fig. 3d). A few cells (15%) are undifferentiated and maintain MPC fate with intermediate amplitude of Hes1 oscillation (Fig. 3e). When illustrated in the 3D "organ" with the interacting network, the BP and PAC fates intermingle with each other. The PACs (red) often emerge centrally in a small cell cluster,

surrounded by BP cells, and BP cells always have at least one PAC neighbor (Fig. 3b, f). PACs are preferentially distributed at the organ surface, where the MPCs initially had mesenchymal neighbors on one side, and thus fewer epithelial contacts (Supplementary Fig. 3d, e). We estimated the number of neighbors that MPCs have in vivo from confocal scans of E10.5 dorsal buds. We found that MPCs located at the surface typically have 3–4 neighbors in a single plane, which we estimate translates to 6–7 neighbors in 3D, while centrally located MPCs have 6–7 neighbors in a single plane (estimated 12–14 neighbors in 3D), similar to the numbers we use in the model. Moreover, in a model variant where all cells are assigned the same number of neighbors (see Methods), the PACs scatter in the organ rather than at the surface

**Fig. 2 | *Trans*-interaction maintains MPC state and *cis*-interaction benefits cell fate segregation. a** 3D maximum intensity projections and pancreatic bud volume quantification of E10.5 *Dll1*^ΔFoxa2^, *Jag1*^ΔFoxa2^, and *Dll1; Jag1*^ΔFoxa2^ embryos compared to littermate controls. Embryos were stained for Pdx1 (green), Ngn3 (red), and Gcg (blue) by whole-mount IF. Scale bar: 50 μm. dp: dorsal pancreas; vp: ventral pancreas. Quantification is shown as mean ± SD, $N = 14$ control-, $N = 7$ Dll1 mutant-, $N = 3$ Jag1 mutant-, $N = 4$ Jag1; Dll1 mutant-embryos. Statistical significance: dp: *Dll1*^ΔFoxa2^: $p < 0.0001$ vs Control; *Jag1*^ΔFoxa2^: $p = 0.0012$ vs Control; *Dll1; Jag1*^ΔFoxa2^: $p < 0.0001$ vs Control, $p < 0.0001$ vs *Jag1*^ΔFoxa2^, $p = 0.5934$ vs *Dll1*^ΔFoxa2^. vp: *Dll1*^ΔFoxa2^: $p = 0.0007$ vs Control; *Jag1*^ΔFoxa2^: $p < 0.0001$ vs Control; *Dll1; Jag1*^ΔFoxa2^: $p = 0.9633$ vs Control, $p < 0.0001$ vs *Jag1*^ΔFoxa2^, $p = 0.0352$ vs *Dll1*^ΔFoxa2^. One-way ANOVA with Tukey's post-hoc test for multiple comparisons. **b** Schematic diagram of the two-cell system and gene expression pattern in different cell fates. **c** Schematic diagram of the *trans*-interaction strength within the two-cell model. With the same intercellular binding activity of receptor and ligand, the cell increases its Hes1 production when *trans*-interaction strength changes from weak to strong. **d** Schematic diagram of the *cis*-interaction strength within the two-cell model. The receptors and ligands are removed with an increased rate when the *cis*-interaction strength changes from weak to strong. **e** Cell fate segregation happens after temporal MPC fate with medium *trans*-interaction. The cells bifurcate into two states with low Dll1: high Hes1 or high Dll1: low Hes1 (left). The transient MPC state characterized by anti-phase oscillations with an intermediate amplitude of Hes1 is indicated with the arrows (right). **f** Cell fate segregation is blocked with strong *trans*-interaction for two cells. Hes1 and Dll1 oscillate inside the cells (left) indicating the cells maintain BP states, and Hes1 oscillates in anti-phase between the BP cells (right). **g** Cell fate segregation is blocked with weak *cis*-interaction for two cells. **h** Cell fate segregation happens bypassing the MPC fate with weak *trans*-interaction. The cells directly bifurcate into low Dll1: high Hes1 and high Dll1: low Hes1 states. **i** Cell fate segregation happens with strong *cis*-interaction. In (**e-i**), balls indicate final cell fates for two cells, red: PAC, green: BP, and brown: MPC; color coded rectangles indicate the strength of *trans*- and *cis*-interaction as shown in (**c**) and (**d**). With corresponding blue or red color, the cells' initial conditions (triangles), dynamic trajectories (lighter curves), and final states (darker curves) are plotted on Dll1-Hes1 plane (left panels) and Hes1 expression is additionally plotted over time in minutes (right panels).

(Supplementary Fig. 3f–h). A positional cue is therefore naturally embedded in the structure in the absence of any other signaling cues since different compositions of epithelial and mesenchymal neighbors contribute different amounts of Dll1 and Jag1 ligands from epithelial neighbors. Thus, cells at the surface, and not their neighbors in the interior of the epithelium, preferentially adopt a PAC fate. The model thus presents a theoretical explanation of how spatial cues in the developing organ contribute to achieving the correct PD distribution of cell fates.

## Modulating Hes1 expression changes cell fate proportion

As the main effector coupling Notch signaling to cell fate, Hes1 is assumed to inhibit the differentiation towards PAC by repressing Ptf1a expression. Perturbations of Notch signaling, and thereby Hes1 expression, are therefore expected to affect the proportions of BP and PAC fates in the pancreas. This is indeed observed in embryos deficient for Hes1 or the ubiquitin ligase Mib1, which is required for *trans*-activation of Notch[33,34]. We tested the ability of our model to recapitulate this shift in cell fate by changing the parameters associated with Hes1 production and comparing the simulation results with experimental observations where Notch activation was chemically perturbed[16].

Theoretically, the Hill constant, $K_2$, quantifies the response of Hes1 to Notch signaling. With a certain level of activated Notch signaling, a large or small $K_2$ implies a weak or strong Hes1 response to the signaling respectively (see Methods, Eq. (1)). Correspondingly, in chemical perturbation experiments, the Hes1 response was weakened or strengthened by the treatments with the γ-secretase inhibitor DAPT or the Nedd8-activating enzyme inhibitor MLN4924, respectively (Fig. 4a). These treatments are assumed to only change the regulation of activated Notch to Hes1 transcription but not the binding affinity of Notch receptor and ligand in the *trans*-interaction. Thus $γ_2$ is not changed in the simulations of treatments.

Weakening Notch-Hes1 activation significantly decreases the amplitude of Hes1 oscillations in the pancreatic cells, and this is seen both in silico and in the experimental data (DAPT vs. DMSO) (Fig. 4b, c). The model predicts that not all cells adopt the PAC fate when Hes1 levels are reduced and thus the inhibition of Ptf1a is released in silico. Instead, many cells fail to differentiate and stay in the MPC state (Fig. 4d, f, Supplementary Movie 2). In contrast, when enhancing Hes1 levels either in silico or by MLN4924 treatment, it results in a high Hes1 oscillation amplitude (Fig. 4b, c), and the BP fate is quickly adopted by most of the in silico cell population (Fig. 4e, g, Supplementary Movie 3).

These predictions agree well with the experimental data from pancreas explants treated with DAPT or MLN4924 (Fig. 4h–j). We obtained quantitative data from a previous publication[16] and re-plotted

these to illustrate how attenuating Hes1 expression with DAPT inhibited adoption of the Sox9^Hi^ BP fate and promoted the Ptf1a^+^ PAC fate. However, PAC fate was only promoted in a subset of cells, while exit from the Sox9/Ptf1a co-expressing MPC fate was hampered in other cells. Conversely, enhancing Hes1 expression with MLN4924 facilitated differentiation to BP fate and suppressed MPC and PAC fates (Fig. 4k–m). Thus, the simulations correctly predict how modulation of Hes1 expression leads to perturbations of cell fate proportions in the pancreas.

## In silico Dll1 and Jag1 deficiency mirrors in vivo mutations

In the two-cell model, Dll1 and Jag1 deficiency have different impact on cell fate segregation (Supplementary Fig. 2). We then next explored the roles of Dll1 and Jag1 in pancreatic development with the in silico organ model. By blocking the expression of Dll1 or Jag1 respectively, we obtain two variants: a *Dll1 deficient* model and a *Jag1 deficient* model (see Methods). Deficiency of Dll1 or Jag1 has different effects on MPC differentiation (Fig. 5a). Reduction of Dll1 expression decreases final MPC proportion (Supplementary Fig. 4a), while reduction of Jag1 increases the proportion (Supplementary Fig. 4b, c).

The *Dll1 deficient* model (corresponding to *Dll1*^ΔFoxa2^) shows that progenitor cells can still bifurcate into two groups, overlapping with the normal PAC and BP cell fates, without Dll1 (Fig. 5b). The organ can self-organize, producing both the local "salt-and-pepper" pattern and the global PD distribution of cell fates (Fig. 5b, Supplementary Movie 4). Conversely, in the *Jag1 deficient* model (corresponding to *Jag1*^ΔFoxa2/−^), the progenitor cells fail to exit the MPC state and maintain intermediate levels of Ptf1a expression and Hes1 oscillation amplitude (Fig. 5c). The virtual organ suffers a severe failure in both cell fate segregation and structural self-organization (Fig. 5c, Supplementary Movie 5). Pairs of interacting cells are bifurcating into the two alternative cell fates in wild-type and *Dll1 deficient* models, while they maintain MPC identity in the *Jag1 deficient* model (Fig. 5d–f). Remarkably, the in silico "phenotypes" produced by the *Dll1 deficient* and *Jag1 deficient* models accurately reflect the in vivo phenotypes of *Dll1* and *Jag1* deficient embryonic pancreata. Compared to wild-type (*R26*^Yfp/+^) and conditional *Dll1* deficient (*Dll1*^ΔFoxa2^) mice, the pancreata of *Jag1* deficient (*Jag1*^ΔFoxa2/−^) mice present a different distribution of cell fates. At day E13.5, the *Dll1*^ΔFoxa2^ pancreas appears identical to the wild-type pancreas with almost complete segregation of MPCs into PAC and BP fates. In contrast, many progenitors maintain an MPC fate in the *Jag1*^ΔFoxa2/−^ pancreas (Fig. 5g–i; ref. [16]).

In the multi-cell model, if Jag1 is absent, the 6-min delay in Dll1 transcription dramatically reduces the amplitude of Hes1 (Supplementary Fig. 4d) but does not cause severe "oscillation death". With Jag1 present, the delay mildly affects the gene expression dynamics

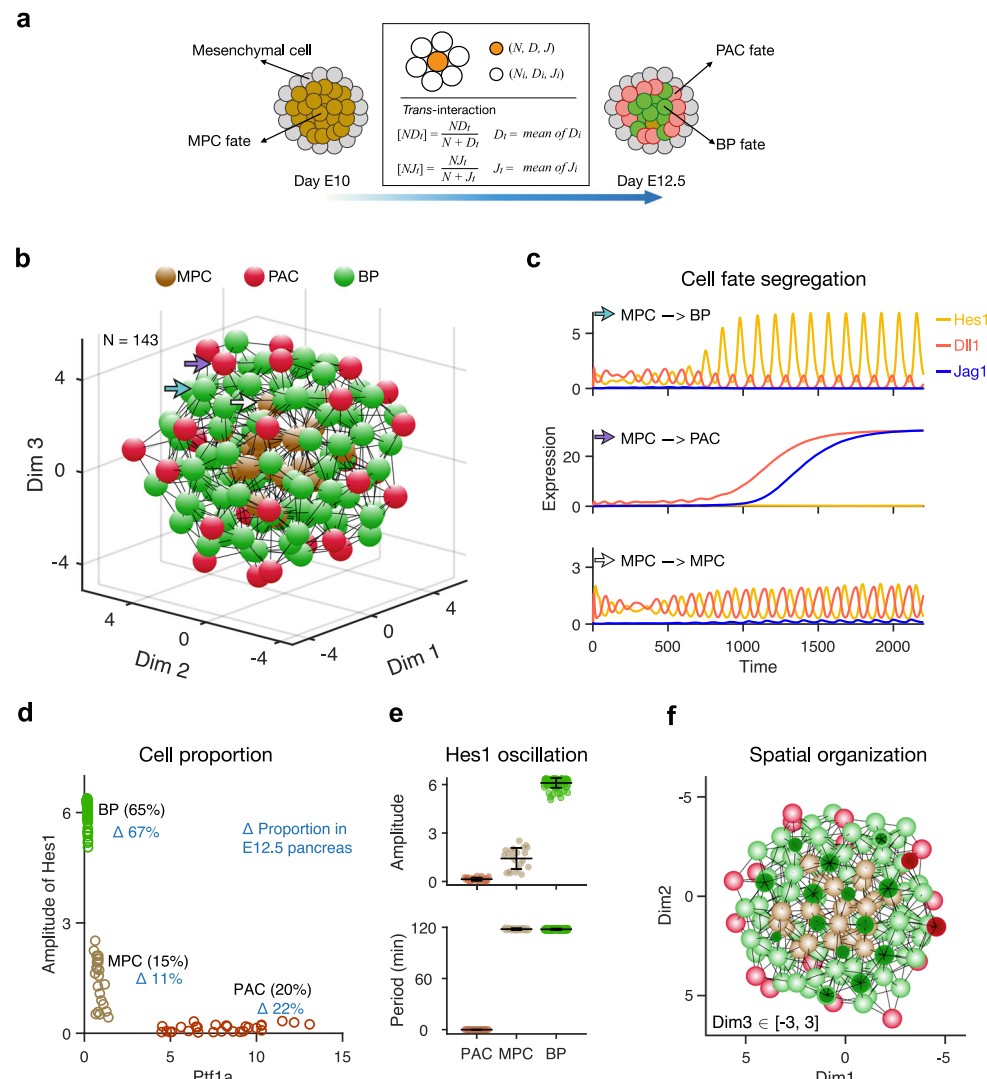

**Fig. 3 | Mathematical model for Notch signaling mediated MPC differentiation.** **a** Schematic diagram of how the simulated cells differentiate from MPC fate to PAC fate or BP fate in a 3D structure during E10-E12.5. Different colors indicate different cell fates hereafter: brown, MPC; red, PAC; green, BP. Cells receive ligands from their neighbors and change their gene expression. **b** E12.5 pancreas in silico. Spatial organization of different cell fates in the simulated pancreatic epithelium at E12.5 is shown with nodes (cells) and edges (interactions). Colors of nodes indicate three different fates: Brown: MPC fate, red: PAC fate, and green: BP fate. **c** Examples of gene expression dynamics when cells differentiate to BP fate (top) and PAC fate (middle) or maintain MPC fate (bottom). Hes1 (yellow), Dll1 (red), and Jag1 (blue) change along the developmental time (min). Hes1 and Dll1 show comparable antiphase oscillation in the same cell when the cell is at MPC state. The positions of the cells are pointed out with corresponding colored arrows in (**b**). The level of Jag1 ($\approx$0.1 $\mu$M) is low but significant at the early time. **d** Comparison of cell proportions in in silico and in vivo E12.5 pancreas. The cells are plotted with the amplitude of Hes1 and Ptf1a level at the final stage. Cell proportions in silico are labeled in black and in vivo are labeled in blue. **e** Statistics of amplitude and period of Hes1 recaptured in the in silico E12.5 pancreas. The amplitude of Hes1 is higher in BP cells than in MPC cells, and the period of Hes1 in BP fate and MPC fate is $\approx$120 min. Mean ± SD, PAC: $N = 28$; MPC: $N = 22$; BP: $N = 93$. **f** Cross-section display of the in silico E12.5 pancreas. The PACs are distributed at the surface of the epithelium surrounded by BP cells, while a few MPCs are in the center closely interacting with the BP cells.

(Supplementary Fig. 4e, f) during cell fate segregation and does not change the final cell proportions (Supplementary Fig. 4g, h). Intriguingly, as the delay increases, the MPC proportion keeps decreasing and the BP cell proportion keeps increasing (Supplementary Fig. 4i). With a 15-min delay, only 3% of the cells were MPCs. This result suggests that time delay in Dll1 transcription might hamper the maintenance of MPC fate in early pancreatic development. Unlike in the two-cell model, "oscillation death" is not observed in the multi-cell model since more neighbors of the cells make it difficult to produce local coherence of Notch signaling.

### Jag1 *cis*-inhibition is crucial for cell fate choice

The two-cell model suggests that Dll1-mediated *trans*-activation helps maintain the MPC fate when Jag1 expression is low at the early stage.

Moreover, in vivo data suggest that Dll1 *trans*-activation is crucial for maintaining normal MPC proliferation and that the low Jag1 levels found in MPCs act via *cis*-inhibition to attenuate proliferation[16]. As Jag1 levels increase they tend to increase the strength of lateral inhibition between neighboring cells and promote the segregation of cell fates.

We therefore next focused on how Jag1 promotes segregation of cell fates. From the two-cell model, we infer that strong Jag1 *trans*-activation and strong *cis*-inhibition (Fig. 2) both contribute to the mechanism by which Jag1 promotes cell fate segregation. The *trans*-activation and Hes1-mediated feedback inhibition of ligand expression drive lateral inhibition to trigger cell fate segregation, and *cis*-inhibition is predicted to strengthen lateral inhibition, but whether *cis*-inhibition is required for efficient cell fate segregation is unknown.

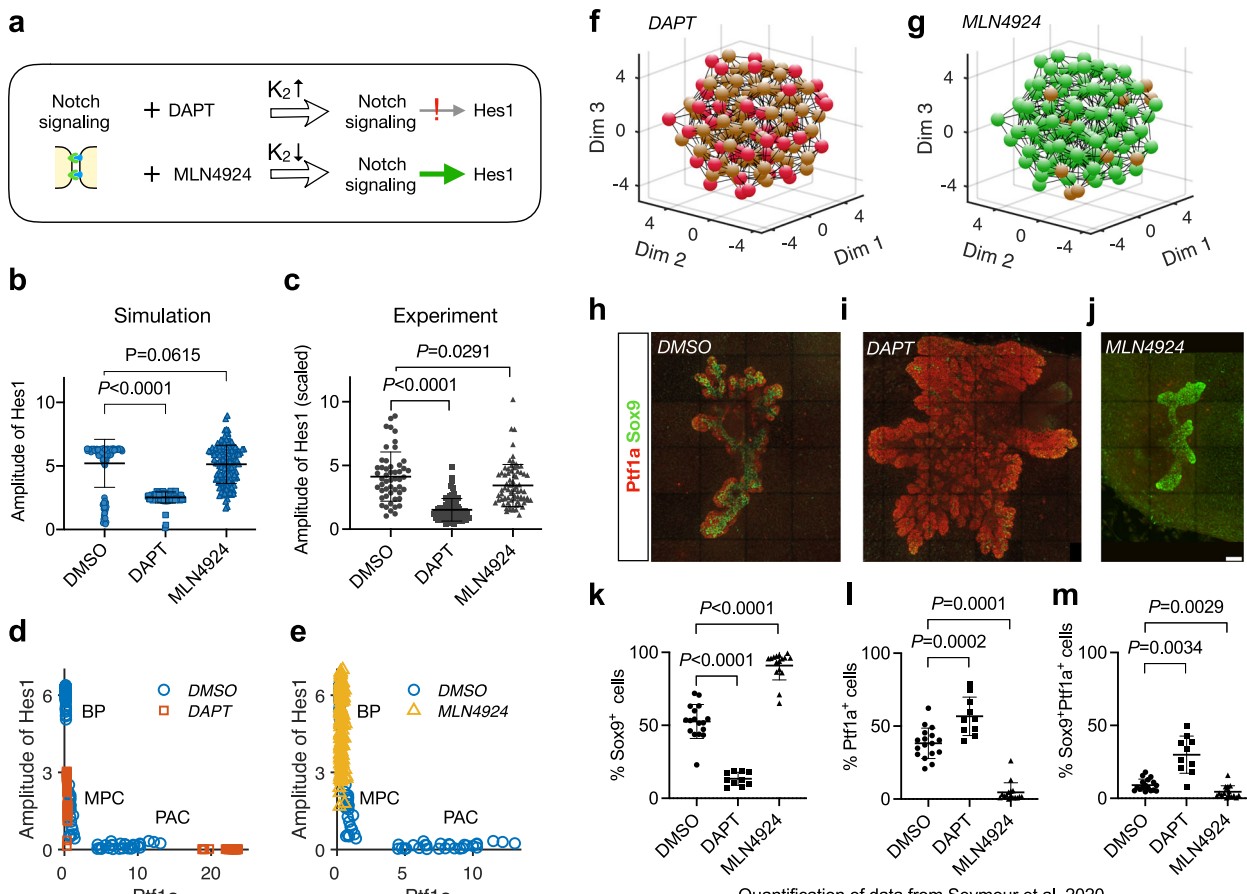

**Fig. 4 | Alteration of Hes1 transcription biases cell proportions. a** Effects of DAPT or MLN4924 on Hes1 expression correspond to increasing or decreasing the parameter $K_2$ in the model. **b** Model predicted effects of treatment with DAPT or MLN4924 on amplitude of Hes1 expression. Values of $K_2$ in different conditions: DMSO ($K_2 = 0.06$), DAPT ($K_2 = 0.3$), and MLN4924 ($K_2 = 0.01$). Mean ± SD, $N = 115$ for DMSO, $N = 91$ for DAPT, and $N = 143$ for MLN4924. $P$ values are calculated with two-tailed Mann–Whitney test and shown above the sample pairs. **c** Experimentally observed relative changes by treatment with DAPT or MLN4924 on the amplitude of Hes1 expression. Data from ref.[16] are plotted (Mean ± SD, $N = 51$ for DMSO, $N = 71$ for DAPT, $N = 74$ for MLN4924), and the original luminescence signal is scaled with $10^4$. $P$ values are calculated with two-tailed Mann–Whitney test and shown above the sample pairs. **d** Amplitude of Hes1 and Ptf1a level changes with DAPT treatment in simulation. Treatment with DAPT is predicted to lead to failure of cell fate segregation and result in more cells maintaining MPC state. In the simulation, 24% of the cells adopt PAC fate, which is slightly higher than DMSO (20%). DMSO:

$N = 143$, DAPT: $N = 143$. **e** Amplitude of Hes1 and Ptf1a level changes with MLN4924 treatment in simulation. Treatment with MLN4924 is predicted to improve the probability of BP cell fate commitment and result in fewer PAC cells in the pancreas. DMSO: $N = 143$, MLN4924: $N = 143$. **f** BP fate is hampered by DAPT treatment in simulation, compared to the normal structure (Fig. 3b). **g** PAC fate is hampered by MLN4924 treatment in simulation, compared to the normal structure (Fig. 3b). **h–j** Cell fate distributions in pancreatic explants with different treatments. Expression of Sox9 (green, BP fate) and Ptf1a (red, PAC fate) by IF is shown as 3D maximum intensity projections in E10.5 pancreas explants after culture for 5 days in DMSO, DAPT, or MLN4924. Scale bar: 100 $\mu$m. **k–m** Statistics of BP, PAC, MPC cell proportions in experiments with different conditions. Data from ref.[16] is plotted (Mean ± SD, $N = 17$ for DMSO, $N = 16$ for MLN4924, $N = 10$ for DAPT). $P$ values are calculated with Two-tailed Welch's $t$ tests and shown above the sample pairs.

Remarkably, the cells of the multi-cell model fail to segregate when *cis*-inhibition is blocked, even when *trans*-interaction is maintained. The likelihood of segregation increases with increasing strength of the *cis*-interaction, and the model predicts a minimal threshold for when cells in the 3D virtual pancreas differentiate properly (Fig. 6a, Supplementary Fig. 5a). These results indicate that *cis*-interaction is a crucial driving force for cell fate bifurcation. Without Jag1, cell fate segregation fails even with higher *cis*-interaction rate (Fig. 6b, Supplementary Fig. 5b). When the *cis*-interaction rate keeps increasing, some of the cells differentiate without Jag1 probably through the expression of Dll1. Conversely, increased expression of Jag1 facilitates cell fate segregation by compensating for the lower *cis*-interaction rate (Fig. 6c, d).

Furthermore, we dissect the roles of *cis*- and *trans*-interaction of Dll1 and Jag1 by removing each of the four interactions (see Methods) in the model. We found that removing *cis*-interaction of Dll1 affects MPC maintenance early on (Supplementary Fig. 5c) and prevents the

cell fate segregation. Removing *trans*-interaction of Dll1 also disturbs the gene expression early on, resulting in high temporal Dll1 expression and earlier cell fate segregation compared with wild type (Supplementary Fig. 5d). Removing *cis*- or *trans*-interaction of Jag1 does not affect MPC maintenance early on (Supplementary Fig. 5e, f). With only *trans*-interaction of Jag1, the cells cannot establish proper cell fate segregation (Supplementary Fig. 5g, h). With only *cis*-interaction of Jag1, the cells can still establish proper cell fate segregation (Supplementary Fig. 5g, h) with a slightly lower amplitude of Hes1. Thus, both *cis*- and *trans*-interaction of Dll1 are responsible for the maintenance of MPC state early on, and the *trans*-interaction of Dll1 also contributes to the maintenance of the MPC state at later on. Jag1 mediates cell fate segregation through *cis*-interaction at a later time (Supplementary Fig. 5i).

To summarize, our model accurately predicts that Dll1 and Jag1 have different roles in pancreatic PD patterning and provides insight into the underlying mechanisms. Dll1 is important for the

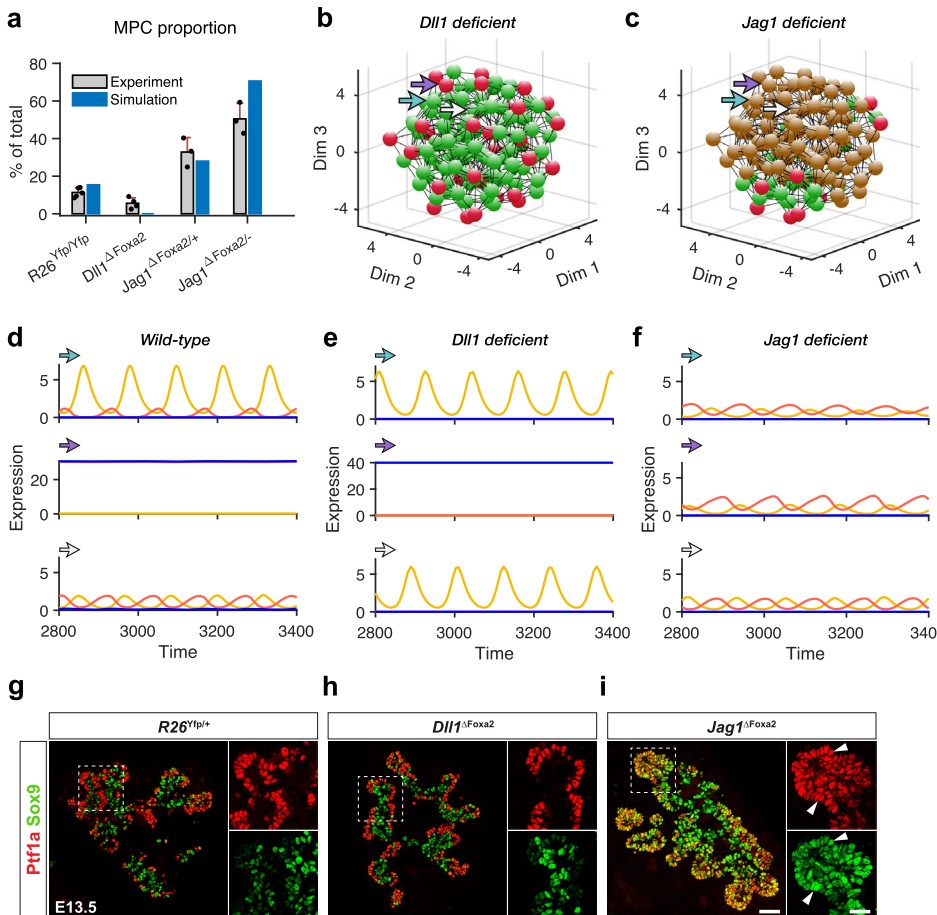

**Fig. 5 | Jag1 plays pivotal role in pancreatic development. a** Dll1 deficiency facilitates while Jag1 deficiency hampers MPC differentiation. Compared with wild type ($R26^{Yfp/Yfp}$), *Dll1* deficient ($Dll1^{\Delta Foxa2}$) mutants show decreased MPC proportion in E12.5 pancreas. Intermediate Jag1 deficiency ($Jag1^{\Delta Foxa2/+}$) and severe Jag1 deficiency ($Jag1^{\Delta Foxa2/-}$) increased MPC proportions in E12.5 pancreas. Gray bars: experimental MPC proportions of pancreas in different mutants, mean ± SD of data in ref. [16] is plotted. $N = 5$ for $R26^{Yfp/Yfp}$ controls, $N = 4$ for Dll1 mutants, $N = 3$ for *Jag1* heterozygous mutants, $N = 3$ for *Jag1* homozygous mutants. Blue bars: predicted MPC proportions of different simulated genotypes with in silico pancreas. **b**, **c** Spatial positions of three cell fates in *Dll1* deficient pancreas and *Jag1* deficient pancreas. Cell fates are defined with the expression of Ptf1a and Hes1 at the final

stage. **d**–**f** Examples for cell fate changes with genotypes. The dynamics of three cells are shown with their expression of Hes1 (yellow), Dll1 (red), and Jag1 (blue). Their positions in the 3D spatial structure are labeled with corresponding colored arrows in (**b**) and (**c**). **g**–**i** Experiments: Distribution of PAC fate and BP fate indicated by cell fate markers, Sox9 (BP, green) and Ptf1a (PAC, red) in $R26^{Yfp/+}$ mice, $Dll1^{\Delta Foxa2}$ mice, and $Jag1^{\Delta Foxa2}$ mice at E13.5. The left panels show the merged channels in each pancreas, and the right panels show the separate channels. When two colors appear in the same cell, the merged color tends to be yellow particularly seen in $Jag1^{\Delta Foxa2}$ mice (arrowheads indicate colocalization). The scale bar for the merged channels is 50 $\mu$m while for the separate channels it is 25 $\mu$m. Representative example of IF staining repeated on three different sets of embryos.

maintenance of the MPC state (Fig. 6e) and Jag1 for cell fate bifurcation (Fig. 6f). We propose that the *cis*-inhibitory action of Jag1 in emerging PACs is as crucial for its ability to facilitate cell fate segregation as its *trans*-activating properties are for specification of proper numbers of BPs.

Increased expression of Jag1 is regulated positively by Ptf1a, which has a relatively slow decay. Thereby the model suggests that Ptf1a serves as a time average of Hes1 expression. It transfers an oscillating Hes1 signal to an average level that determines Jag1 expression levels. Since it is directly inhibited by Hes1, Dll1 oscillates in anti-phase with Hes1 when the Ptf1a is low or moderate. As a consequence, Dll1 has less of an effect on differentiation.

The transcription factor Ptf1a governs the delay of Jag1 relative to Dll1. This in turn decides the duration of the MPC state (Supplementary Fig. 6a–c). A slower Ptf1a implies a slower differentiation. If Jag1 is regulated directly by Hes1 (Supplementary Fig. 6d), which is equivalent to a infinitely fast Ptf1a response to Hes1, the time window before the differentiation of MPCs becomes very short (Supplementary Fig. 6e, f), which indicates no pancreas development since the tissue cannot have proper MPC expansion. Thus, Ptf1a is critical as a buffering and

averaging element in pancreatic development. In addition to Ptf1a, we observed that the degradation time of free notch receptor ($\tau_n$) also affects the timing of cell fate segregation but not cell proportions (Supplementary Fig. 6g–i).

The rationales of how Notch signaling mediates cell fate differentiation are vindicated by the theoretical model's robustness to the selection of parameter values (Supplementary Fig. 7). The oscillation period of Hes1 is central in the differentiation and is partially influenced by the system it is embedded in. In addition to the amplitude, the period of Hes1 oscillation was also affected under some conditions. For example, the period was increased by MLN4924 treatment[16]. In the model, a few parameters can affect the period when they change (Supplementary Fig. 8). Consistent with MLN4924 treatment, smaller $K_2$ leads to a longer period of ≈160 min (Supplementary Fig. 8a). The auto-inhibition time delay of Hes1 ($\tau_0$) induces a continuous change of the period when it varies. Each variable has at least one relevant parameter that can change the period, indicating the period is a result of the coupling of gene expression dynamics in the system.

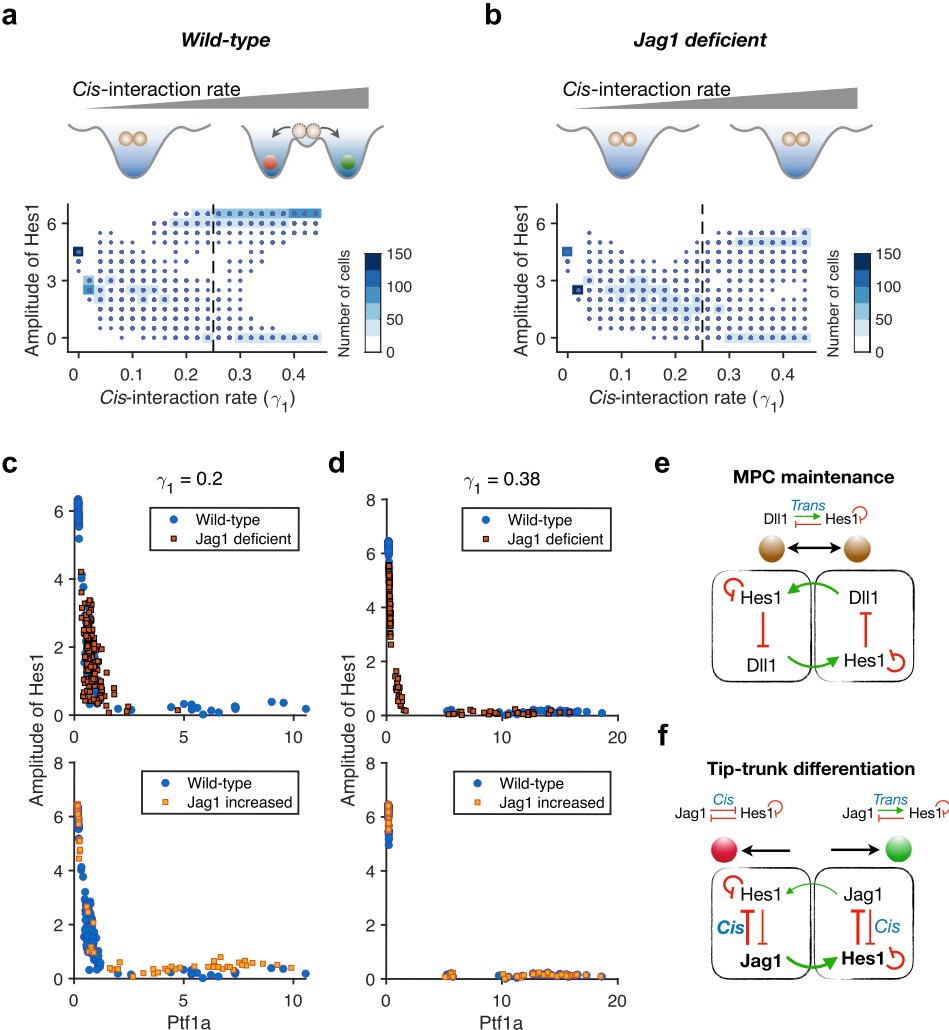

**Fig. 6 | Jag1 bifurcates cell fates by mediating strong *cis*-interaction. a** Summary schematic diagram for the necessity of *cis*-interaction in pancreatic cell fate segregation. Successful cell fate bifurcation requires a certain level of *cis*-interaction. The heatmap shows the distribution of amplitude of Hes1 in simulated pancreas with different *cis*-interaction rate. Each column corresponds to one specific value of the *cis*-interaction rate. Cells maintain intermediate Hes1 oscillation amplitudes with low *cis*-interaction, while either high or low Hes1 oscillation amplitudes are seen with increased *cis*-interaction. **b** Summary schematic diagram for cell fate segregation failure without Jag1 regardless of the *cis*-interaction rate. With high *cis*-interaction rate (0.4), cell fate segregation is only partially achieved and some cells still fail to exit the MPC state and maintain an intermediate amplitude of Hes1. **c** Model predicts that increased expression of Jag1 rescues cell fate segregation at low *cis*-interaction rate. With $\gamma_1 = 0.2$, a few cells in wild type (blue dot, $a_J = 1.0$) pancreas can differentiate into BP fate or PAC fate, with Jag1 deficiency (red square, $a_J = 0$), the cells maintain MPC fate and with increased Jag1 (orange square, $a_J = 4.0$), the cells adopt a BP or PAC fate except for very few cells that maintain an MPC fate. **d** Model predicts that expression of Jag1 facilitates cell fate segregation at high *cis*-interaction rate. With $\gamma_1 = 0.38$, Jag1 deficiency causes a few cells to fail to differentiate and maintain MPC fate. With increased Jag1 ($a_J = 1.2$ is shown), the cells adopt a BP or PAC fate like the wild type. **e** MPC state is maintained by low Jag1 expression at the beginning of pancreatic development. The *trans*-interaction between cells and Hes1 oscillation are key factors for MPC fate maintenance. **f** Tip-trunk differentiation happens when Jag1 rises and mediates strong *cis*-interaction. Cell fate symmetry is broken by *cis*-interaction and strengthened *trans*-interaction.

## Discussion

Precise timing and positional control of cell fate choices are fundamental for organ formation in development. Here we present a mathematical model that accurately recapitulates pancreatic proximodistal (PD) patterning of early MPCs. The model allows us to dissect the mechanisms of spatiotemporal self-organization of the PAC and BP fates during in silico pancreatic development.

By including two different Notch ligands, Dll1 and Jag1, with subtle differences in their gene regulatory network connections and by incorporating cell-autonomous *cis*-inhibition as well as *trans*-activation mediated by cell-cell interactions in a multi-cell model, a group of pancreatic cells will reliably transit from the MPC state to the two mutually exclusive but coexisting BP and PAC fates with the correct spatial arrangement. Dll1-mediated *trans*-activation of Notch receptors drives Hes1 expression, which by virtue of Hes1 auto-inhibition, is

oscillatory in the MPC state. The oscillatory dynamics prevent interacting MPCs from being forced into different states where one of the cells in an interacting pair would always be driven to differentiate by the lateral inhibition mechanism, and maintaining adequate numbers of MPCs would thus be difficult. A similar oscillatory mechanism for the mutual maintenance of the progenitor state between neighboring progenitor cells has been suggested to operate in the central nervous system[35,36]. Later, the slowly responsive Ptf1a provides an average readout of the fluctuating Hes1 level and activates the other Notch ligand, Jag1. Jag1 increases *cis*-inhibition and amplifies the mutually inhibitory effects of the lateral inhibition feedback. The strengthened lateral inhibition breaks the symmetry of MPC maintenance and leads to cell fate bifurcation.

This way of achieving symmetry breaking is different from that used in neuronal progenitors of the dorsal forebrain. Here, progenitors

are also maintained by oscillatory expression of Dll1 and Hes1, and the transcription factor that activates Dll1 expression in their associated GRN is Neurogenin2 (Ngn2, ref. [35]). Low, oscillating levels of Ngn2 and its target gene Dll1 activate Hes1 in neighboring cells where Hes1 cyclically represses Ngn2 and Dll1 as well as itself to maintain oscillatory dynamics and progenitor fate. Again, symmetry is broken when Hes1 is downregulated and Ngn2 and Dll1 therefore reach stable high levels that induce neuronal differentiation. However, Hes1 downregulation is not due to Jag1-mediated *cis*-inhibition as Jag1 and Dll1 are expressed in different non-overlapping domains in the central nervous system[37,38]. Instead, it might be achieved by the action of miR-9, whose mature form accumulates over time and eventually causes degradation of *Hes1* mRNA[39,40].

The cells of our multi-cell model have different numbers of epithelial and mesenchymal neighbors depending on whether they are located at the periphery or in the central parts of the structure. Since only epithelial cells are contributing ligand input to neighboring cells, this architecture naturally gives rise to a bias when the cell fates bifurcate into PAC and BP fates. Cells located inside the structure have more epithelial neighbors contributing Notch ligands and tend to become BP cells, while cells on the epithelial surface preferentially become PACs since they have mesenchymal neighbors on one side, which are assumed to provide no ligand input. How the initial "salt-and-pepper"-like pattern distributes along the PD axis thus becomes "predictable" by the composition of neighbors. In reality, pattern formation is also influenced by other processes including cell sorting due to differential adhesion[17]. Also, since mesenchymally produced FGF10 is required to maintain epithelial Ptf1a expression[41], another contributing factor may be the different concentrations of FGF10 reaching the epithelial cells depending on their distance from the mesenchyme. This would be increasingly important as the pancreas anlage gets bigger over time. However, at early stages where the pancreatic bud is composed of a few hundred cells, this is likely of minor importance. This early stage is recapitulated by our model and our results indicate that positional cues affect cell fate determination via differential Notch signaling in the early phase of pancreatic development. This early pattern could then be reinforced by a gradient of FGF10 as the organ grows.

Dll1 and Jag1 play different roles in pancreas development and knockout of the two ligands have very different phenotypic consequences[16]. Our model recapitulates these phenotypes and strongly suggests that these differences are caused by the slow intermediate regulatory protein Ptf1a which acts as a gatekeeper for differentiation. The Dll1 gene is directly repressed by Hes1 and indirectly through Hes1 control of Ptf1a expression, while Jag1 is only regulated through the expression of Ptf1a. This results in oscillatory Dll1 expression and more stable Jag1 expression, which again results in different levels of *cis*-inhibitory and *trans*-activating input to Notch receptors.

In agreement with experiments, our model implies that reducing or blocking Dll1 still allows PD patterning. Further, it shows robustness against increased Jag1 activity. Overall, our analysis suggests that *cis*-interaction by Jag1 is the driving force for cell fate bifurcation and PD patterning. Consistent with this notion, we show here that the *Dll1;Jag1* double mutant bud volumes are unchanged or slightly increased compared to *Dll1* single mutants, arguing that *cis*-inhibition is the most parsimonious explanation for how Jag1 inhibits Notch activity at the early bud stage. One confounding factor for this interpretation could be residual *trans*-activation from another ligand. However, the only other ligand expressed in the pancreas is Dll4, which is found in Ngn3+ endocrine precursors[26,27,42]. Due to the low number of Dll4+Ngn3+ cells relative to Dll1+Sox9+Ptf1a+ MPCs, we assume that Dll4 only makes a minor contribution to the Notch *trans*-activation in the E10.5 MPCs. We are currently studying Dll4-deficient embryos to test this notion. Moreover, it should be noted that even though our model assumes the same affinity of Dll1 and Jag1 for *cis*-binding, a Jag1-specific role for *cis*-inhibition relies on Jag1 having a higher affinity than Dll1 for Notch, a notion that is consistent with the absence of fringe expression at E10.5[43].

*Cis*-inhibition has also been found to strengthen the lateral inhibition between mutually exclusive cell fates in the patterning of the developing sensory organ in *Drosophila*[44]. Notably, in silico elimination of *cis*-inhibition resulted in the maintenance of the MPC state, similar to what is observed in vivo in Jag1 deficient pancreas. However, MPC symmetry is broken eventually, resulting in the excessive formation of PACs around E14.5[16]. The reason for this prominent shift from MPC to PAC fate is presently unknown, but it was speculated that onset of Lunatic fringe expression around E14.5 in the distal part of the epithelium[43] might play a role. On a larger perspective, beyond the Notch-Delta system, then cell fate choices are sometimes found to be dependent on cell-intrinsic feedback[5,44–47]. However, within paradigms that are governed by Notch-Delta systems, our model suggests a dependence on the cell-intrinsic feedback and illustrates how it plays out in early pancreas differentiation. Our extended Jag1 model also goes beyond the standard "salt-and-pepper" pattern associated with Notch-Delta signaling and demonstrates how a simple positional cue can modulate Notch activity to achieve an ordered PD pattern in the pancreas.

## Methods

### Mice and Immunofluorescence Imaging

All mice were housed at the Department for Experimental Medicine at the University of Copenhagen and breeding and experimental work was approved by "Miljø−og fødevarelseministeriet−Dyreforsøgstilsynet". All animals were housed with a standard 12 h light:12 h dark cycle and were provided with standard laboratory chow and water ad libitum. Ambient temperature was $22 \pm 2\,°C$ and $55 \pm 10\%$ humidity. All animal experiments described herein were conducted in accordance with local legislation and authorized by the local regulatory authorities. Embryos were dissected at noon ± 1 h at the gestational age indicated and noon of the day the plug was observed was set to be E0.5. Where necessary, tissue from limb buds, head or tail tip was used for genotyping as were ear biopsies from weaned mice of the various lines. DNA was extracted using Quick-Extract I DNA Extraction Solution (Epicentre). Mouse lines are described in ref. [16] and genotyping was performed as described there. Embryos or isolated foreguts were fixed in 4% paraformaldehyde in PBS at 4 °C for 45 min to 3 h and cryopreserved in 30% sucrose in PBS overnight (O/N) at 4 °C, equilibrated in Tissue-Tek O.C.T. for 1 h at room temperature (RT) then embedded in O.C.T. on dry ice slabs. Frozen sections were cut at $10\,\mu m$ on a Thermo Scientific Microm HM 560 cryostat and mounted on Superfrost Plus slides (Thermo Fisher Scientific). For detection of Notch ligands, IF analysis was commenced the day of sectioning after 1 h air-drying sections at RT. For storage, slides were frozen at −80 °C with silica desiccant (Merck; Cat#103804). IF analysis was performed on a minimum of 3 embryos per genotype unless otherwise stated. Following air-drying or thawing (for 5-10 min) of frozen cryosections, slides were washed for 10 min in PBS then subjected to antigen retrieval in pH 6.0 citrate buffer for 1 h at 37 °C. After 3 washes in PBS, slides were permeabilized in 0.15% Triton X-100 in PBS for 1 h at RT. They were then blocked in 1% normal donkey serum in PBS with 0.1% Tween-20 (PBST) for 2−5 h at RT then were incubated O/N at 4 °C with primary antibodies diluted in the same buffer. Slides were subsequently washed 3 times in PBS then primary antibodies were detected with donkey-raised anti-rabbit, -guinea pig, -mouse, -rat, -goat, -sheep or -chicken and goat-raised anti-Armenian hamster secondary antibodies conjugated to either Cy5 (1:500), Cy3 (1:1000), Alexa Fluor 488 (1:1000) or DyLight 405 (1:200) (all Jackson ImmunoResearch) in a 2 h RT incubation. Primary antibodies are listed in Supplementary Table 1.

E10.5 whole embryos destined for WM-IF analysis were fixed in buffered 4% formaldehyde (VWR/Prolabo) O/N at 4 °C then washed 3 times in PBS before being dehydrated through an ascending series of MeOH concentrations in PBS. Tissues were equilibrated in 100% MeOH on ice for 1 h before being stored in fresh 100% MeOH at −20 °C. WM-IF was performed as follows: Specimens were incubated in Dent's bleach (MeOH:DMSO:30% H$_2$O$_2$, 4:1:1) O/N at RT then washed in MeOH before being equilibrated to PBS through a descending series of MeOH concentrations in PBS. Tissues were then blocked O/N at RT in 0.5% TNB (a proprietary TSA-block supplied in the TSA Cyanine 3 System Kit from Perkin Elmer), then incubated for 48 h at 4 °C with primary antibodies diluted in the same buffer. Tissues were subsequently washed extensively in PBS at RT before being incubated 48 h at 4 °C with donkey-raised secondary antibodies conjugated to Alexa Fluor 488/Cy2, Cy3, or Cy5 (all Jackson ImmunoResearch) all diluted 1:500 in 0.5% TNB. Finally, specimens were washed at RT in several changes of PBS before equilibration to 100% MeOH through an ascending series of MeOH concentrations in PBS and imaged immediately or stored at 4 °C. For imaging, tissues were cleared and mounted in BABB (benzyl alcohol:benzyl benzoate, 1:2) in a coverslipped glass concavity slide.

IF-stained frozen sections were imaged on a Zeiss LSM780 confocal microscope using Plan-Apochromat 40x/1.3 Oil or C-Apochromat 40x/1.2 objectives or on a Leica SP8 confocal microscope using an HC PL APO CS2 40x/1.30 OIL objective at between 512 x 512 or 2048 x 2048 resolution and 1x or 2x zoom factor. Tile scans were stitched immediately within the Leica LAS AF software. BABB-cleared WM-IF specimens were scanned with a Zeiss LSM780 confocal microscope using a W N-Achroplan ×20/0.5 objective.

## Bioluminescence Imaging and Analysis

Bioluminescence data were obtained from ref. [16]. Briefly, explants were imaged on the stage of an inverted Olympus IX83 microscope, maintained at 37 °C in 5% CO$_2$. 1 mM Beetle Luciferin (Potassium Salt, Promega; Cat#E1601) was added to the medium before imaging. The bioluminescence signal was collected on an Olympus IX83 microscope using a UPLFLN 40X/1.3 oil immersion objective and a cooled iXon Ultra 888 EMCCD camera (Oxford Instruments, Andor). The exposure time was 10 min/frame with no binning. Image sequences were analyzed in Fiji (ImageJ version 2.0.0, NIH) as described in detail in ref. [16]. To track single cells the path was defined with the ROI tool and supported by a Maximum Projection image. A custom plug-in, Z-axis Profiler Plus[36], was used to extract the bioluminescence signal per cell over time for peak-to-peak quantification in Microsoft Excel. Mean amplitude for each oscillating cell was calculated as the difference in relative luciferase activity (a.u.) between individual peaks (P) and troughs (T) using the equation: Mean amplitude = [(P1−T1) + (P2−T2) + ⋯ + (P$n$−T$n$)]/$n$. Image sequences covering 20 h were assembled into movies in Fiji. The bioluminescence signal from each cell was time-aligned as indicated for a single cell in Fig. 1h in order to align every measured cell in the heatmaps shown in Fig. 1g.

## Cell and pancreatic bud volume quantification

Jag1, Dll1, and Hes1 co-expression in Ptf1a and Nkx6-1 expressing compartments were quantified in Adobe Photoshop from quadruple Jag1, Dll1, Ptf1a, Nkx6-1 IF stainings and triple Hes1, Ptf1a, Nkx6-1 IF stainings, respectively. Where required for multiplex IF, antibodies were eluted in 25 mM glycine, 1% SDS (pH 2) for 1 h at 60 °C as described in ref. [48] and modified by ref. [49]. For E10.5 co-expression cell counts 383 cells from four embryos were counted for Jag1, 512 cells from five embryos for Dll1, and 159 and 157 cells from two embryos were counted for Nkx6-1 and Hes1, respectively. A total of 1,470 cells from three E12.5 embryos were counted for the ligand quantification and 1168 cells from three E12.5 embryos were counted for Hes1 quantification. Quantification of E10.5 bud volumes was based on Pdx1 immunoreactivity in z-stacks using Imaris v9.5 (BitPlane). The Surface-

rendering-area function was used with "detail" set to 2 $\mu$m in conjunction with the "smooth" function in Imaris. Manual masking was used to isolate the pancreatic buds.

## Western blot and protein half-life determination

For protein half-life, 266-6 cells (ATCC CRL-2151) were grown overnight in DMEM (Gibco). The identity of the cell line was verified less than one year ago by IF analysis of pancreatic marker expression, which showed it to be Sox9$^+$, Nkx6-1$^+$, Pdx1$^+$ as expected. At time $t = 0$, 100 $\mu$M cycloheximide (CHX) was added to prevent further protein synthesis. At the indicated time points, cells were lysed for 10 min in ice-cold RIPA buffer with 1x phosphatase inhibitor cocktail (Sigma-Aldrich) and Complete Ultra Protease inhibitor (Thermo Scientific/Roche). Cell lysates were sonicated 5 × 30 s ON/OFF on a Diagenode BioRuptor in 1.5 mL eppendorff tubes followed by centrifugation at 21,000 $g$ for 30 min at 4 °C and stored for Western blot analysis. Pierce BCA protein kit (ThermoFisher) was used to measure protein concentration on a Nanodrop 2000 (ThermoFisher). Western blot analysis: Lysates were boiled for 5 min in Laemmli sample buffer and 10 $\mu$g protein was separated by electrophoresis on NuPage 4–12% BisTris SDS-PAGE gels in MOPS buffer (ThermoFisher) and transferred to PVDF membranes (Bio-Rad) using the Bio-Rad Mini-Protean transfer system. Membranes were blocked in SuperBlock (ThermoFisher) for 1 h at RT and incubated with mouse anti-pan-Actin (MA5-11869, Thermo Scientific), goat anti-Jag1 (sc-6011; Santa Cruz), and rabbit anti-Ptf1a[50] primary antibodies. After three washes with TBS-T (0.1% Tween-20 in 1x Tris-buffered saline), the blot was incubated with the respective species-specific secondary HRP-conjugated antibodies at RT for 30 min. ECL Prime Western Blotting Detection Reagent (Sigma-Aldrich) was used for detection according to the manufacturer's instructions. Protein degradation half-lives were calculated using first-order rate kinetics. Data collected from different time points were plotted with exponential curve fitting using robust regression in Prism v9.4.1 (GraphPad) and half-life was calculated by ln2/exponent of curve function.

## 3D interaction structure for cell-cell communication

To simulate the cells' interaction with their neighbors, we formed a three-dimensional structure with an expanded method previously described in ref. [51]. Regarding the cells' current locations, the cell$_j$ was defined as a neighbor of cell$_i$ if cell$_j$ had shortest distance to the midpoint of these two cells compared with any other cells. If cell$_j$ was a neighbor, it was assumed to have direct contact with cell$_i$. In the multi-cell model, 400 cells were simulated as interacting particles. They started from random locations and moved to their final locations based on the repulsive and attractive forces acting between the cells. Specifically, we iterated the cells' location regarding the potential for pairwise interaction between two interacting cells as

$$V = e^{-r} - e^{-r/5},$$

where $r$ is the distance between the two cells. Each cell's movement is dependent on the integrated potential from all its neighbors during each iteration step. Enough iteration steps were taken until the cell centers did not move anymore (100 iterations in this study).

A static three-dimensional interaction network of the cells was constructed from the final locations. In the static spatial structure, the convex hull and all the neighbors of its nodes were set to be the mesenchymal cells surrounding the epithelial MPC cells. Among the 400 cells, 257 cells were mesenchymal cells, and 143 cells were epithelial cells. These 257 mesenchymal cells did not give Notch signaling in the simulation. Each cell sensed the ligands from its neighbors which were constrained by the interactions. In the experiments, the two-dimensional confocal scans show that the central MPCs of E10.5 dorsal buds have 6–7 neighbors. Assuming the cells are randomly packed soft

balls, the cells should have a similar number of neighbors on the orthogonal plane of the observed plane. Thus, the pancreatic cells have 12–14 interacting neighbors, which is consistent between the three-dimensional structure model and the real pancreas.

To elucidate the boundary effects on cell fate distribution, another three-dimensional interaction network was derived with the constraint that all the cells had the same number of neighbors. In that model, only the epithelial cells were considered, and for each cell the 12 closest epithelial cells were defined as neighbors.

## Gene expression dynamics in each cell

We considered the expression dynamics of Hes1, Dll1, Jag1, Notch, and Ptf1a on the basis of the simplified gene regulatory network (Fig. 1i). The equations described the time variations in the concentrations of these variables with the following notations: Hes1 ($H$), Dll1 ($D$), Jag1 ($J$), Notch ($N$), and Ptf1a ($P$).

For each cell, we assumed that the Notch ligand and receptor at the membrane are evenly distributed and proportional to the concentrations. The *cis*- and *trans*-interaction were dependent on the amount of ligand and receptor and constant rates. For simplicity, we ignored the direct binding competition between *cis*- and *trans*-interaction. The *cis*-interaction was calculated as

$$\gamma_1[ND] = \gamma_1 \frac{ND}{N+D} \text{ and } \gamma_1[NJ] = \gamma_1 \frac{NJ}{N+J}$$

The terms are derived based on the assumption that the receptor and ligand bind tightly. Within only the process of binding, if current concentrations of notch receptor and Dll1 ligand are $N$ and $D$, the concentration of complex is $[ND]$, we have

$$N = N^{\text{total}} = N^{\text{free}} + [ND] \text{ and } D = D^{\text{total}} = D^{\text{free}} + [ND].$$

On the other hand, at the steady state,

$$[ND] = \frac{1}{K_D} N^{\text{free}} D^{\text{free}}, K_D \text{ is the dissociation constant of the binding.}$$

Such that,

$$[ND] = \frac{1}{K_D} (N^{\text{total}} - [ND])(D^{\text{total}} - [ND]).$$

with $[ND] \ll \min\{N^{\text{total}}, D^{\text{total}}\}$, the term $[ND]^2$ could be omitted,

$$[ND] = \frac{1}{K_D} (N - [ND])(D - [ND]) \rightarrow [ND] \approx \frac{ND}{K_D + N + D}.$$

$K_D \approx 0$ for tight binding, and the binding process is fast compared with the gene expression. Thus, when wired in the equations for gene regulations, the *cis*-interaction could be approximately represented with an additional limited rate $\gamma_1$ as

$$\gamma_1[ND] = \gamma_1 \frac{ND}{N+D}.$$

This mathematical approximate method is used for all *cis*-interactions and *trans*-interactions between Notch receptor and Dll1/Jag1 ligands. The simplified calculation of $[ND]$ is applicable when considering the reality that $D \ll N$ in pancreatic cells. This simplification may not be applicable for the systems where the concentrations of receptors and ligands are comparable.

The terms avoid overestimation of binding activity when the receptor or the ligand is saturated for the binding. With receptor ≫ ligand, increasing receptor does not increase the binding activity

infinitely. Similarly, with ligand ≫ receptor, increasing ligand does not increase the binding activity infinitely.

In the *trans*-interaction, the intercellular ligand surrounding the cell was approximately calculated as the mean concentration of all its neighbors,

$$D_t = \sum_{i=1}^{i=n} D_i/n \text{ and } J_t = \sum_{i=1}^{i=n} J_i/n, n \text{ is the number of neighbours for the cell.}$$

The activated Notch signaling thus depends on the interactions with all its neighbors. If the neighbors are mesenchymal cells, $D_i = J_i = N_i = 0$. Thereby cells on the surface have less Notch signaling.

With free Notch receptor $N$, available ligands from its neighbors $D_t$ and $J_t$, the binding activity of *trans*-interaction was calculated as

$$\gamma_2[ND_t] = \gamma_2 \frac{ND_t}{N+D_t} \text{ and } \gamma_2[NJ_t] = \gamma_2 \frac{NJ_t}{N+J_t}$$

In a similar way, we can get the removed amount of Dll1 and Jag1 through *trans*-interaction as

$$\gamma_2[DN_t] = \gamma_2 \frac{DN_t}{D+N_t} \text{ and } \gamma_2[JN_t] = \gamma_2 \frac{JN_t}{J+N_t}$$

In this network motif, the regulation input from one gene to another was formulated by Hill functions with modest Hill coefficients up to 2. We did not consider a difference in binding affinity or other factors between Dll1 and Jag1 when binding to Notch receptors. To keep the model simple and reduce parameters to as few as possible, we set the Hill constant from Ptf1a to Dll1 and Jag1 to the same determined by $K_4$ and $K_6$. The *trans*-interaction results in producing the transcriptionally active NICD, which functions as transcription activator of Hes1[10]. The NICD is transiently present on the promoter and rapidly turned over through ubiquitin-mediated protein degradation[52–55]. Fast dynamics of NICD leads to sensitive activation of Hes1 transcription by Notch signaling. Thus, here we simply assume that the NICD $= K_c\gamma_2([ND_t] + [NJ_t])$. The contribution of *trans*-interaction to Hes1 production through NICD was integrated as

$$\frac{\gamma_2^2([ND_t] + [NJ_t])^2}{\gamma_2^2([ND_t] + [NJ_t])^2 + K_2^2}, \text{where } K_c \text{ is absorbed in } K_2, K_2 \propto 1/K_c.$$

With above assumptions, we mathematically described the gene regulatory network motif with:

$$\frac{dH}{dt} = a_H \frac{K_1^2}{K_1^2 + H(t-\tau_0)^2} \frac{\gamma_2^2([ND_t]+[NJ_t])^2}{\gamma_2^2([ND_t]+[NJ_t])^2 + K_2^2} - \frac{H}{\tau_h} \quad (1)$$

$$\frac{dD}{dt} = a_D \frac{K_3^2}{K_3^2 + H^2} + a_w \frac{P^2}{P^2 + K_6^2} - \gamma_1[ND] - \gamma_2[DN_t] - \frac{D}{\tau_d} \quad (2)$$

$$\frac{dJ}{dt} = a_J \frac{P^2}{P^2 + K_4^2} - \gamma_1[NJ] - \gamma_2[JN_t] - \frac{J}{\tau_j} \quad (3)$$

$$\frac{dN}{dt} = a_N - \gamma_1[ND] - \gamma_1[NJ] - \gamma_2[ND_t] - \gamma_2[NJ_t] - \frac{N}{\tau_n} \quad (4)$$

$$\frac{dP}{dt} = a_P \frac{K_5^2}{K_5^2 + H^2} - \frac{P}{\tau_p} \quad (5)$$

**Table 1 | Parameters (italic) used in the simulation of gene expression dynamics**

| Parameters | Description | Value in 3D model (two-cell model) |
|---|---|---|
| $a_H$ | Maximum production rate of Hes1 | 3.0 (5.0) $\mu$M min$^{-1}$ |
| $a_D$ | Maximum production rate of Dll1 regulated by Hes1 | 0.5 $\mu$M min$^{-1}$ |
| $a_w$ | Maximum production rate of Dll1 regulated by Ptf1a | 0.8 (1.0) $\mu$M min$^{-1}$ |
| $a_J$ | Maximum production rate of Jag1 | 1.0 $\mu$M min$^{-1}$ |
| $a_N$ | Maximum production rate of Notch | 0.5 $\mu$M min$^{-1}$ |
| $a_P$ | Maximum production rate of Ptf1a | 0.2 (0.1) $\mu$M min$^{-1}$ |
| $\tau_0$ | Time delay for Hes1 auto-inhibition | 40 min[30] |
| $\tau_h$ | Degradation time of Hes1 | 20 min[30] |
| $\tau_d$ | Degradation time of Dll1 | 50 min[32] |
| $\tau_j$ | Degradation time of Jag1 | 120 min[24], Supplementary Fig. 1 |
| $\tau_n$ | Degradation time of Notch | 50 min |
| $\tau_p$ | Degradation time of Ptf1a | 120 min[23], Supplementary Fig. 1 |
| $\gamma_1$ | Degradation rate by cis-interaction | 0.25 $\mu$M min$^{-1}$ |
| $\gamma_2$ | Degradation rate by trans-interaction | 0.02 (0.1) $\mu$M min$^{-1}$ |
| $K_1$ | Hill constant for Hes1 auto-inhibition | 0.3 (0.5) $\mu$M |
| $K_2$ | Hill constant for Notch signaling activating Hes1 | 0.06 (0.5) $\mu$M |
| $K_3$ | Hill constant for Hes1 inhibiting Dll1 | 0.8 (1.0) $\mu$M |
| $K_4$ | Hill constant for Ptf1a activating Jag1 | 4.0 $\mu$M |
| $K_5$ | Hill constant for Hes1 inhibiting Ptf1a | 0.1 (0.2) $\mu$M |
| $K_6$ | Hill constant for Ptf1a activating Dll1 | 4.0 (10.0) $\mu$M |

If the level of the NICD is added to the model as a variable:

$$\frac{d[\text{NICD}]}{dt} = a_{[\text{NICD}]}\gamma_2([ND_t] + [NJ_t]) - \frac{[\text{NICD}]}{\tau_{[\text{NICD}]}}, \quad (6)$$

the simulation results of cell fate differentiation do not change with proper selection of parameters.

With fast degradation of NICD ($\tau_{[\text{NICD}]} \approx 10$ min), the dynamics of gene expression are similar to the simplified description. With slow degradation of NICD ($\tau_{[\text{NICD}]} \approx 45$ min), the anti-phase oscillations shift to become in-phase at the transient MPC state (in two-cell model). Importantly, Bray and colleagues measured the half-life of *Drosophila* NICD after co-immunoprecipitation with Su(H) to be ≈10 min[56]. We therefore believe that the simplified description is a valid approximation.

The biological meaning of parameters and values used in these equations are listed in Table 1.

We use similar values of the parameters measured by previous studies and this study. For the unknown parameters we choose the values which can explain the experimental results. We systematically analyzed the sensitivity of each parameter with the two-cell model. The model is robust to selection of parameters. Some other parameters could affect cell fate segregation in addition to $\gamma_1$ (Supplementary Fig. 7). The period of Hes1 could also change when some of the parameters change (Supplementary Fig. 8).

### Simulation of phenotypes and conditions

Identical initial conditions for the simulations with the two cells are $[H(0), D(0), J(0), N(0), P(0)] = [0.5, 1.1, 0, 1, 0]$ and $[0.6, 1, 0, 1, 0]$. In 3D models, the initial conditions of cells at time $t = 0$ were all set to $[H(0), D(0), J(0), N(0), P(0)] = [0.1, 0.1, 0.1, 0.1, 0.1]$, which represents the start from a low level of Hes1, Dll1, Jag1, Notch, and Ptf1a. The models were solved by a standard MATLAB (R2019a, 64-bit) delay differential equations solver (dde23).

We performed the simulation with the time interval up to 3600 min (2.5 days). For wild-type simulation, we used the parameters in Table 1. *Dll1 deficient* mutant was simulated with $a_D = 0$ and $a_w = 0$, and *Jag1 deficient* mutant was simulated with $a_J = 0$. To simulate *type2*

*Dll1* mutant with delay in Dll1 transcription, Eq. (2) was modified as

$$\frac{dD}{dt} = a_D \frac{K_3^2}{K_3^2 + H(t-\tau)^2} + a_w \frac{P(t-\tau)^2}{P(t-\tau)^2 + K_6^2} - \gamma_1[ND] - \gamma_2[DN_t] - \frac{D}{\tau_d}.$$
$$(7)$$

The parameter $\tau = 6$ min was used when simulating the *type2 Dll1* mutant as indicated by experimental measurement. The effect of different $\tau$ was also studied in final cell fate proportions.

The small-molecule treatments perturbed NICD levels in cells by inhibiting NICD production (DAPT) or inhibiting NICD degradation (MLN4924), indicating that $K_c$ is smaller or larger with the same level intercellular interaction of receptor and ligand ($\gamma_2$). Thus $K_2$ was increased to simulate the effect of DAPT treatment and was decreased to simulate the effect of MLN4924 treatment.

The boundary effects are induced by different proportions of epithelial and mesenchymal neighbors among cells. To clarify the tendency of PACs' distribution at the epithelial surface is due to the surface cells having more mesenchymal neighbors and fewer epithelial cells, a model with the same number of neighbors for each cell is formed by forcing each cell to have 12 closest cells from the 143 epithelial cells as their neighbors. The same parameter was used except a slightly larger $K_2 = 0.08$ and small noise in the initial condition to ensure all the cells adopt either BP fate or PAC fate.

To dissect the roles of *cis-* and *trans-*interaction of each ligand, the representative terms for each type of interaction in the equations were removed respectively. Specifically, the term $\gamma_1[ND]$ was removed to simulate pancreatic development without *cis-*interaction of Dll1, and the terms $\gamma_2[ND_t]$ and $\gamma_2[DN_t]$ were removed for simulation without *trans-*interaction of Dll1. A similar method was applied to Jag1.

To illuminate how the dynamic of Ptf1a impacted the cell fate segregation, a parameter $\delta$ was embedded in Eq. (5):

$$\frac{dP}{dt} = (a_P \frac{K_5^2}{K_5^2 + H^2} - \frac{P}{\tau_p}) \cdot \delta \quad (8)$$

$\delta < 1$ indicates a slower time scale and $\delta > 1$ indicates a faster time scale of Ptf1a compared to wild type ($\delta = 1$). These methods enable us to tune

the time scale of Ptf1a without changing the steady state level of Ptf1a. When $\delta \gg 1$, Ptf1a responds to Hes1 instantly, which is equivalent to Hes1 regulating Dll1 and Jag1 directly.

## Statistics

To define the final fate of each cell in the simulations, we quantified a few variables to describe the state of the cells based on the gene expression in the late time interval from 3200 min to 3600 min. The Ptf1a expression was calculated as the average of the respective solutions in this time interval.

To calculate the amplitude of Hes1, the peaks and valleys for each Hes1 track were initially identified. The amplitude was estimated as the difference between the mean values of the peaks and the mean value of the troughs. The periods were calculated as the mean time difference between two adjacent peaks. Mann–Whitney tests were applied to compare the Hes1 amplitudes between treatments and control samples.

Statistical significance of differences in pancreatic bud volumes between genotypes was assessed by one-way ANOVA followed by Tukey's post-hoc test for multiple comparisons using Graphpad Prism v9.4.1.

## Reporting summary

Further information on research design is available in the Nature Portfolio Reporting Summary linked to this article.

## Data availability

The datasets generated during and/or analyzed during the current study are presented in the paper. The Western blot, cell counting, and oscillation quantification data generated in this study are provided in the Supplementary Information/Source Data files. Source data are provided with this paper.

## Code availability

The code generated during and/or analyzed during the current study is available at https://doi.org/10.5281/zenodo.7071389[57].

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

## Acknowledgements

The authors thank the DanStem and reNEW Imaging Core and Jutta Bulkescher for technical assistance. K.S. was supported by the European Research Council (ERC) under the European Union's Horizon 2020 research and innovation program under grant agreement No 740704. M.H.J. was supported by the Independent Research Fund Denmark (grant number: 9040-00116B) and the Novo Nordisk Foundation (grant number: NNF20OC0064978). A.T. was supported by the Danish National Research Foundation through StemPhys Center of Excellence (grant number: DNRF116). P.S. was supported by the Independent Research Fund Denmark (grant number: 9039-00232B) and the Novo Nordisk Foundation (grant numbers: NNF18OC0034136 and NNF20OC0063628). The Novo Nordisk Foundation Center for Stem Cell Biology and Novo Nordisk Foundation Center for Stem Cell Medicine were supported by Novo Nordisk Foundation grants NNF17CC0027852 and NNF21CC0073729, respectively.

## Author contributions

X.X., P.Ser., M.H.J., K.S., A.T. designed the study. All the authors performed the research and wrote the manuscript. X.X. implemented models and all the simulations. P.Sey., A.R.E., M.C.J., and P.Ser. did the experiments. All authors reviewed the manuscript.

## Competing interests

The authors declare no competing interests.
