## [Peer Review File · Nature Communications]

Jag1-Notch cis-interaction determines cell fate segregation in pancreatic developmentREVIEWER COMMENTS

Reviewer #1 (Remarks to the Author):

Xu 2021 review report

The manuscript by Xu et al focuses on developing a mathematical model for the differentiation of multipotent progenitor cells (MPCs) to unipotent pro-acinar cells (PACs) and bipotent duct/endocrine progenitor cells (BPs). The paper aims at developing a model that reproduces the experimental observations about the different role of Jag1 and Dll1 in the first stage of pancreatic cells differentiation as observed by an earlier paper (ref 14). The main modeling approach uses delayed autoinhibition feedback of Hes1 (accounting for Hes1 oscillations) combined with Notch mediated lateral inhibition (accounting for differentiation into two distinct fates). The model describes a gene regulatory network where the Notch ligands Jag1 and Dll1 are wired in a different manner, which according to the authors is the main reason for the different roles of the two Notch ligands as well as the different phenotypes generated by mutating either Dll1 and Jag1. Finally, the authors consider the roles of cis-inhibition between ligands and receptors, and the role of cluster geometry on differentiation dynamics into the two distinct states.

The general concept of modeling Hes1 oscillations combined with lateral inhibition is an interesting one, and is certainly needed in the field. The model indeed captures some of the main experimental features such as the initial Hes1 oscillations and the resolving to two distinct states. However, there seems to be some major problems with the implementation of the model as well as with some of the conclusions drawn from it. In particular, one of the main claims is that the difference between the phenotypes of Dll1 and Jag1 stems from the direct regulation of Dll1 by Hes1 vs indirect regulation Jag1 by Hes1 (Fig. 1F). As detailed below (point 1), it seems that within the parameters of the model Dll1 does not do much (i.e. taking it out Dll1 does not affect Hes1 oscillations), This seems to make some of the analysis irrelevant or trivial and seems to contradict some of the experimental results in ref 14. In addition, there are technical issues with the way some of the mutant were described. Also, to support some of the claims the authors should compare alternative models with different assumptions. Overall, it seems to me that the model and analysis is interesting but there needs to be some major modifications and additional analysis in order to support the claims.

A detailed list of major Issues:

1. One of the main points brought by the authors is the difference in the regulation of Dll1 and Jag1 within the GRN (Fig. 1F) as the main reason for the diverging phenotypes of Dll1 and Jag1 deletions. However, it seems that with the chosen parameters, the regulatory branch of Dll1 is not doing much in the model. This can be seen in two places: First, in Fig. 5B,E the Dll1 deficient model does not show any effect on Hes1 oscillations (namely, Dll1 branch does not affect Hes1). Second, in Fig. S1C, Hes1 dynamics seems to be almost independent of Dll1 (but they do depend on Jag1 in Fig. S1D). It seems that the parameters controlling Dll1 (a_D and a_W) seem to be too weak compared to the parameters regulating the Jag1 to actually make a difference. If that is the case, then the fact that model 'predicts' different behaviors for Dll1 and Jag1 mutants is rather trivial – Dll1 doesn't really do anything. This contradicts Fig. 6G claiming Dll1-Hes1 feedback is important for MPC. As far as I can tell, in ref 14 it is shown that modulating the transcriptional delay of Dll1 generate observable phenotypes (affect Hes oscillations), which means that Dll1 is not obsolete in this stage. The authors should try to modify their model so that the Dll1 branch is relevant, and hopefully reproduce the effects observed experimentally. If the authors think that indeed the Dll1 branch does not do much, it should be omitted from the model altogether.

2. I think there are a few technical problems with the definition of the model and its implementation that need to be clarified:

2a Implementation of Dll1 deletion. To implement the deletion the authors set a_D to 0 (line 381, methods). I think they need to set both a_D and a_W to 0, since in the deletion there should be no expression of Dll1 at all.

2b In line 370 the authors use unusual normalization for interaction term (e.g. $[ND_t] = (ND_t)/(N+D_t)$). It should be explained why this normalization is used for a biochemical reaction.

Why not use standard biochemical equilibrium expression $[ND_t] = (ND_t)/k_t$ as is typically being used?

2c In line 375 (model equations) – In the Delta equation, why are the activation by P and repression by H are additive and not multiplicative (as in the equation on H). This creates a situation where even at very high H levels, there is no complete suppression. Please justify this assumption.

3. Another main point in the manuscript is that Hes1 indirectly regulates Jag1 through Ptf1 (compared to direct regulation of Dll1). According to the manuscript this is important since Ptf1 has a slow decay rates and serves as a time average of Hes1 expression. To really show this point the authors need to compare 3 models: (i) The model they suggest. (ii) A model with fast Ptf1 decay rate. (iii) A model where Hes1 directly regulates Jag1 (as it does for Dll1). Comparing these models will help the authors support their claim regarding the role of Ptf1 as a buffering and averaging element.

4. The claims in Fig. 6 and Fig. S4 are unclear. First, Figs. 6B-C only show what happens in WT with high cis-inhibition but not with low cis-inhibition. It is unclear what Fig. S4 is trying to show. The authors should show Figs. 6B-C without cis-inhibition (or how it changes with cis-inhibition strength). Fig. 6E-F show that in Jag1 deletion, there is no real lateral inhibition. This is not surprising given the issue raised in point 1 above (since it's the main branch through which interactions occur).

5. The authors assume the same strength of cis-interaction for Dll1 and Jag1. This is likely not the case as it has been shown that strength of cis interactions may be very different between Dll and Jag and may be modulated by fringe (see for example LeBon et al eLife 2014). It could be that the different phenotypes associated with Dll1 and Jag1 may be related to this property and not to the difference in GRN. Can the authors test that option (Or at least argue whether this should be important or not)?

6. Parameter choice – The authors do not explain how they chose the parameters for the “WT” model. I think they should add a short explanation that answers the following questions: Are the chosen parameter values based on experiments? Are they fine tuned to match experimental results? If so, how sensitive are the results to a small change in each parameter?

7. The time delay for Hes1 chosen in the model is 40 mins. The oscillations in the model (for example Fig. 2) seem to have a period close to the observed 90 mins. Is the difference coming from the coupling to Notch? How does the oscillation period depends on the time delay?

8. For the 3D model, the authors need to show statistics indicating if the results are simply because of edge cells having less inhibition from their neighbors? This can be tested by considering geometry with periodic boundary conditions. Is Hes1 in the actual system show lower expression at boundary cells?

Detailed list of minor issues:

9. In the abstract, line10, it seems more appropriate to replace “The model predict...” with “the model captures”

10. In the abstract, line 15-16 the claim is that trans-activation feedback is associated with Dll1, but this is not manifested in the model. (see major point 1)

11. On line 38 the authors claim that “ligand degradation is facilitated by cis-interactions”. Is there an experimental evidence supporting this claim?

12. In line 113 the wrong figure is referenced, should be “Figure 1F”.

13. In line 118-119 the explanation about the different simulations should be more accurate. There is no simulation with strong trans-interactions and weak cis-interactions, those are 2 simulations, one with strong trans-interactions and medium cis-interactions and one with medium trans-interactions and weak cis-interactions (at least this is what is shown in figure 2).

14. In line 126 a discussion about what does “strong” or “weak” cis- or trans-interactions mean in terms of model parameters is required. Is there some criteria describing what strong and weak means?

15. In line 128 the word “which” should be deleted (typo).

16. In line 150 it should be clearly stated that it is about the 2-cells model.

17. In line 158 the authors say that in the simulation results PACs have high Ptf1a expression and low Hes1 expression, compared to BPs. This should be quantified.

18. In line 163 a comparison between the number of neighbors of each cell in the simulation and the real tissue should be added. Can the number of neighbors reach 14 (as in the simulation)?

19. In line 192 authors say that changing K_2 corresponds to DAPT and MLN4924, explain why this is assumed.
20. In line 225 explain how can one see, by examining the model results, that "Dll1-mediated trans-activation helps maintain the MPC fate when Jag1 expression is low at the early stage". (This is related to major point 1 above)
21. In line 360, explain how a neighbor is defined in the simulation.
22. In line 364 replace "Figure 1H" by "Figure 1F" (typo).
23. In line 375 (model equations) – A different notation is used for the same thing in different equations, sometimes using the $[\cdot]$ notation for normalized terms and sometimes writing them explicitly.
24. In table1 the description of the half time parameters is wrong (remove "Notch and" from all, looks like a typo).
25. In figure 3, sub-figure D bottom graph – It looks like the PAC result is missing.
26. In figure 4 mark K-M as experimental results in a clear way.

Reviewer #2 (Remarks to the Author):

The manuscript entitled "Jag1-Notch cis-interaction determines cell fate segregation in pancreatic development" by Xiaochan Xu et al. presents an elegant study of cell fate choice mediated by Notch signaling in the pancreas. The study focuses on the choice from multipotent progenitor cells (MPC) to pro-acinar cells (PAC) and duct/endocrine progenitor cells (BP). The results mostly come from modeling (simulation results). The study is based on experimental data previously published by the laboratory of Palle Serup, who is author of the manuscript reviewed here, and provides few new data.

The results present a model for Notch signaling arising from ligands Dll1 and Jag1. The simulation results recapitulate the in vivo expressions and suggest that cis-inhibition mediated by Jag1 is crucial for cell fate choice. The authors set into the model that Hes1 oscillates through a delayed negative feedback, as previously suggested, and drives oscillations of Dll1 by repressing it. Jag1 and Dll1 trans-activate and cis-inhibit Notch signaling. Jag1 does not oscillate because it is regulated by Ptf1a, which has a slow decay and therefore is a slowly evolving variable that can average Hes1 oscillations. The formulated model seems correct based on the hypotheses and knowledge so far and the obtained results seem reasonable.

In my opinion the manuscript provides a plausible model for this developmental process. It represents an effort to formulate a framework capable of reproducing the observations. And this effort is achieved. The model reproduces the expressions of the selected genes in the wild type and the change in proportions of cell fates when Notch signaling is altered. Taken together, the study shows a plausible model for how MPC to PAC/BP fate choice occurs. From this plausible model, the authors extract new knowledge: that Jag1 cis-inhibition is crucial.

However, there is no data to validate the model and this prediction. The same data that are used to construct the model are the ones that the model reproduces (i.e. expression data in the wild type and changes upon Notch alteration, which sustain that Notch and lateral inhibition is relevant). In my opinion, validation of the model requires: A) to test whether additional factors that need to be assumed in the model are really happening in vivo or not (or to prove this is the only framework that works (what is an impossible task)) and B) to test its predictions (i.e the crucial role of Jag1 cis-inhibition). Thus, at present, the model is plausible but is not tested. In this sense, I believe that the manuscript does not provide enough new findings, since it does not validate its proposals, or some of them.

Here below I indicate my specific major concerns:

- 1) In the model, there is a clear asymmetry between Jag1 and Dll1. Jag1 is controlled by Ptf1a while

Dll1 is mainly controlled by Hes1. Because Ptf1a is a slowly evolving variable, Jag1 does not oscillate, whereas Dll1 does. This asymmetry explains why simulations that are deficient in Jag1 drives distinct results than those that are deficient in Dll1. But:

1.a. What are the experimental evidences for the slow decay of Ptf1a? This slow decay is crucial to obtain the results presented in the manuscript. To validate the model it is necessary (among other things) to provide evidence for this slow decay.

1.b. The model sets that Ptf1a regulates weakly Dll1 while it strongly regulates Jag1: what is the evidence in favour of this difference? Supporting this would also provide a partial validation of the model. (In addition, references that support that Hes1 represses Ptf1a should be provided.)

2) In the model there is another asymmetry: γ_1 , which is the rate of (degradation by) cis-interaction, is much larger than γ_2 , which is the rate of (degradation by) trans-interactions.

2.a. What is the evidence for this strong difference? Assessing this will provide a partial validation of the model.

2.b. In addition, while γ_1 is considered to be the rate or strength of cis-interactions (I agree with that) and it is changed to evaluate the effect of cis-inhibition, γ_2 is not changed when evaluating the effect of trans-interactions. Instead, K_2 or a_H are modified to evaluate the effect trans-interactions. In my opinion, a change in trans-interactions should involve a change in two parameters: in γ_2 and in a_H (or K_2). The authors should repeat their analysis of trans-interactions and cis-inhibition, by changing both γ_2 (and a_H or K_2) and γ_1 . This is relevant since the results claim that without cis-interactions (i.e. $\gamma_1=0$), despite changing trans-interactions (through K_2), there is no cell fate choice of BP and PAC. However, without cis-interactions, what happens if γ_2 parameter is as large as γ_1 default value?

2.c. A relevant validation would be to test whether cis-inhibition of Jag1 is indeed relevant or not. I understand that experimentally eliminating cis-inhibition is out of the scope. But what would be expected in Jag1 gain-of-function scenarios in the model? And in Dll1 gain-of-function scenarios? In these gain-of-function scenarios the effect of cis-inhibition and of trans-interactions could be tested in the model, and see whether they raise distinct predictions. Ultimately, this could be validated with data in gain of function embryos.

3) The expression of MPC, PAC and BP cells needs to be clarified:

3.a. At stage 10.5 cells MPC cells have Ptf1a and Hes1 oscillates (line 90 and Figure 1A-B,D). They also have Jag1 (Figure 1A). However, in the simulations, MPC cells do not have Ptf1a nor Jag1 while they have Hes1 oscillating. How are these differences reconciled?

3.b. The model relies on the assumptions that MPC involve oscillations of Hes1 and Dll1 whereas PAC and BP fates correspond to sustained and distinct expressions: PAC cells express Dll1, Jag1 and Ptf1a, whereas Hes1 is expressed in BP cells. In Figure 1C, while I see that Hes1 and Ptf1a are expressed commonly in distinct cells and Jag1 co-localizes with Ptf1a, I do not see more co-localization of Dll1 with Ptf1a than with Hes1, or at least it is not obvious to me. Therefore, I do not see clearly that PAC cells have high Dll1 whereas BP cells do not. Could this be made clearer or clarified?

4) The results show proximodistal (PD) patterning, however, this needs to be further investigated. The authors indicate that the position of cells (being at the surface or not of the epithelium) is a relevant cue for lateral inhibition since cells at the surface interact with less cells than those at the interior. This positional cue drives cells at the surface to preferentially become PAC cells, compared to cells at the interior, and this results in PD patterning. This seems reasonable and to be expected from lateral inhibition as the authors clearly explain. However, the simulations (e.g. Figure 3C) have few cells at the interior, such that most of them have an adjacent cell which is at the surface. Therefore, the PAC cells that arise in the simulation are all or almost all at the surface. Therefore, a clear PD patterning is

found. However, if the simulations had many more cells at the interior, such that their neighboring cells are also all at the interior, then I expect PAC cells to arise at the interior as well, and not only at the surface. Thus, if many more cells are simulated, I expect that the bias of PAC cells being found at the surface will be much less relevant (because cells at the surface will tend to become PAC cells but some cells at the interior will also become PAC cells). In this situation, the PD patterning would be much less apparent, and it may be thought to be a weak cue to account for the in vivo PD patterning (which is much more strong: with a majority of the cells at the surface being PAC). Thus, the authors should justify the number of cells used at the simulations and should run their simulations with higher numbers since as evidenced in Figs.2C,4H, embryos have many more cells than those at the simulations.

Minor comments:

- 1) In Figure 2: the action of cis-inhibition at very early times seems to necessarily occur through Dll1 and not Jag1 since Jag1 is not expressed at early times (although experiments do not clearly support that, e.g. Fig.1). Please discuss and clarify if cis-inhibition of Dll1 or Jag1 is relevant at this early times (that of Dll1 and that of Jag1 could be removed). Depicting Jag1, Dll1 and Ptf1a would also help to clarify.
- 2) To model inhibition of Notch signaling pathway by Nedd8-activating enzyme inhibitor MLN4924: the authors could change τ_n , which is the parameter for Notch degradation without binding to the ligand. Please discuss.
- 3) The sentence "Taken together, the results generated by our models fill a gap in the discussion about how cell-intrinsic feedback can be crucial for cell fate choice, a feature that is absent in most of the theoretical models of Notch signaling." should be made more precise. The effect of cell-intrinsic feedback mediated by cis-inhibition can be seen in Ref [5], Formosa-Jordan et al. Plos one 9, e95744 (2014), Corson et al. Science 356, eaai7407 (2017), Bocci et al. Front Physiol 11:929 (2020), among others.
- 4) In the model, while Jag1 deficient has no Jag1, Dll1 deficient has little Dll1 through a_w which is not set to zero. It would be better to have $a_w=0$ in this deficient scenario, or to justify otherwise.
- 5) Panels D and H of Figure 2 are exactly the same, if I am not wrong. Please indicate so, or just keep only one of them.
- 6) I have not been able to find the files of Supplementary Movies 1-3.
- 7) What is meant by "sorted cells" in Figure S2E?
- 8) Line 169: Not clear the meaning that "there is an initial salt and pepper pattern that later develops to a proximodistal patterning." If I understood correctly, the simulations do not show re-arrangement of cell fates and hence there is not an initial pattern that after a while is re-organized proximally in a distinct manner. Please rewrite the sentence to clarify what the simulations show and what is thought to occur in the embryo.
- 9) Why Sox^{Hi} and Jag⁺ notation and not simply "⁺" or "^{Hi}" (not both notations)? In addition, in the abstract it is used "Jag^{Hi}PAC" and it remains unclear what is meant for.
- 10) Figure 4H-J are from data in ref.[14]. Also Figure 5G-I. Please indicate.
- 11) Figure S3: not clear to me that DAPT, when modeled as an increased K_2 , drives MPC fate or just a PAC with lower Hes1 amplitude (since Ptf1 is very low in these cells). Not clear also which is the proportion of PAC cells compared to DMSO. Please justify and clarify.

12) Please correct errors in the definition of tau_n and tau_p in the Table of parameter values.

13) Introduction: Lines 34-44: more references for the findings that are mentioned are needed. What reference supports that Hes1 represses Jag1 through Ptf1?

14) Why data on oscillations of Hes1 are reanalyzed? What do we learn? Lines 98-106 state this re-analysis but it is unclear what it is useful for in this study.

15) Lines 89-90: add reference and cite Figure 1A.

Reviewer #3 (Remarks to the Author):

In this report by Xu et al authors examine the differentiation of pancreatic multipotent progenitors (MPCs) into pro-acinar cells (PACs) and bi-potential ducto-endocrine progenitors (BPs) using mathematical modeling that couples what is known about dynamic Notch signaling in these cells with the spatial distribution of interacting cells. The authors back up these predictions with experimentation, using both genetics and pharmacological approaches. This study tackles the challenging question of how the first major fate restriction happens in the pancreatic epithelium, that of acinar versus ducto-endocrine. The authors hypothesize that the salt-and-pepper distribution of progenitors could be governed by Notch-ligand mediated lateral inhibition. Furthermore, they incorporate cell-autonomous cis-inhibition and trans-activation mediated by cell-cell interactions in their model, which together predicts how cells with restricted potential (acinar vs ducto-endocrine) sort out into the correct spatial locations (acinar at the tips in the pancreatic periphery vs ducto-endocrine at the core of the pancreatic bud). An interesting prediction is based on the recent finding that downstream of Notch, Hes1 displays oscillatory expression in pancreatic progenitors and this the quality of this oscillation drives MPC and BP fate. Change in oscillation frequency, modulated by slow responsive Ptf1a leads to Jag1 activation and cell fate bifurcation. Similar fate restrictions based on analogous GRNs in the nervous system provides a road map for this study.

This study tackles a difficult question regarding cell fate within the early pancreatic epithelium. The work is based on observations in a paper by some of the authors in Dev Cell in 2020. The manuscript is dense, however, and difficult to get through, as the dynamic interactions are inherently complex. But the methods are not always described clearly, which makes further evaluation difficult. The authors do a valiant effort to explain the ideas using schematics, which is helpful. But more clarification is needed. The authors should consider some the following points.

Major points:

1. It is still unclear to me how the authors can definitively point to the presence of a cis- versus a trans- Notch activation in any particular cell within the early pancreatic bud. They need to make this crystal clear. In addition, they don't consider possible dynamics of the Notch receptors, only using Hes as a proxy. This seems an important omission.
2. Quantification of expression overlaps between Hes, Dll, Jag, Ptf1a, Nkx6.1 is needed from E10.5-12.5.
3. Better explanation is needed for methods in Fig.1D (page 6, line 100). Is this analysis of individual cells done on immunofluorescent stained sections? How many cells/sections/embryos? How are the embryos staged? There is mention of bioluminescence and immunofluorescence. Methods are very unclear as is.

Smaller points:

1. No need for "Experimental.." in the figure title for Fig.1. Just "Expression.." would be better.
2. It is difficult to appreciate some points made in the introduction, where Figure 1A is mentioned. This panel shows Jag1 staining, while the text refers to Dll1.
3. It is difficult to fully appreciate the expression of Jag1/Dll1 in Ptf1a or Hes1 expressing cells without a membrane marker.

point-by-point response to reviewers' comments

We appreciate the reviewers' efforts to evaluate our manuscript. We found the comments and suggestions inspiring and we believe addressing these has improved the paper significantly. The main improvements of our manuscript are briefly summarized below

1. Per Reviewer 1 request, we have investigated the consequences of Dll1 deficiency and Dll1 transcriptional delay in our model to further support our interpretation of the role of Dll1 in pancreatic cell fate segregation.
 - 1) The Dll1 is proved to be important for MPC state maintenance by the consistent results from both experiment and model: Knockdown or knockout Dll1 decreases the proportion of MPC in E12.5 pancreas.
 - 2) We have included new simulations with delay in Dll1 transcription. The model overall can capture the "oscillation death" induced by the delay when Jag1 is absent.
2. Per reviewer's 2 request the size of the silico 3D pancreas is increased from 200 to 400 cells with 143 interior cells, both comparable with the in vivo cell numbers at E10.
3. We have added the sensitivity analysis for all the parameters.
4. New results about how the time scale of Ptf1a affects the cell fate segregation. The faster the time scale of Ptf1a dynamic is, the earlier the cell fate segregation happens.

REVIEWER COMMENTS

Reviewer #1 (Remarks to the Author):

Xu 2021 review report

The manuscript by Xu et al focuses on developing a mathematical model for the differentiation of multipotent progenitor cells (MPCs) to unipotent pro-acinar cells (PACs) and bipotent duct/endocrine progenitor cells (BPs). The paper aims at developing a model that reproduces the experimental observations about the different role of Jag1 and Dll1 in the first stage of pancreatic cells differentiation as observed by an earlier paper (ref 14). The main modeling approach uses delayed autoinhibition feedback of Hes1 (accounting for Hes1 oscillations) combined with Notch mediated lateral inhibition (accounting for differentiation into two distinct fates). The model describes a gene regulatory network where the Notch ligands Jag1 and Dll1 are wired in a different manner, which according to the authors is the main reason for the different roles of the two Notch ligands as well as the different phenotypes generated by mutating either Dll1 and Jag1. Finally, the authors consider the roles of cis-inhibition between ligands and receptors, and the role of cluster geometry on differentiation dynamics into the two distinct states.

The general concept of modeling Hes1 oscillations combined with lateral inhibition is an interesting one, and is certainly needed in the field. The model indeed captures some of the main experimental features such as the initial Hes1 oscillations and the resolving to two distinct states.

We are glad that reviewer finds our manuscript interesting and appreciates the importance of our approach for the field.

However, there seems to be some major problems with the implementation of the model as well as with some of the conclusions drawn from it. In particular, one of the main claims is that the difference between the phenotypes of Dll1 and Jag1 stems from the direct regulation of Dll1 by Hes1 vs indirect regulation Jag1 by Hes1 (Fig. 1F). As detailed below (point 1), it seems that within the parameters of the model Dll1 does not do much (i.e. taking it out Dll1 does not affect Hes1 oscillations), This seems to make some of the analysis irrelevant or trivial and seems to contradict some of the experimental results in ref 14.

- We appreciate reviewer bringing up this point. We can see how from the previous version of the manuscript one could easily arrive to this conclusion. This is however not the case and Dll1 did have a phenotype, which became more pronounced when we increased system size to match that in vivo.
- We have also validated that our model is able to reproduce results in ref 14.
- We are addressing these points in detail below in reply to Point 1 and have extensively revised the manuscript and the corresponding figures to clarify this point.

In addition, there are technical issues with the way some of the mutant were described. Also, to support some of the claims the authors should compare alternative models with different assumptions. Overall, it seems to me that the model and analysis is interesting but there needs to be some major modifications and additional analysis in order to support the claims.

A detailed list of major Issues:

1. One of the main points brought by the authors is the difference in the regulation of Dll1 and Jag1 within the GRN (Fig. 1F) as the main reason for the diverging phenotypes of Dll1 and Jag1 deletions. However, it seems that with the chosen parameters, the regulatory branch of Dll1 is not doing much in the model. This can be seen in two places: First, in Fig. 5B,E the Dll1 deficient model does not show any effect on Hes1 oscillations (namely, Dll1 branch does not affect Hes1).

- As we mentioned above, Dll1 did have a phenotype. While we have not explicitly mention this in the text, removing Dll1 did change Hes1 expression and the resulting cells fate: the MPC cell in Figure 2C (Wildtype) becomes PAC cell in Figure 5B (Dll1 deficient). However, because in the previous version we were simulating only 43 interior cells, this was the only cell affected and thus there were not observable changed in cell proportions. In the updated version, where we increased cell number and simulated 143 interior cells to better match in vivo case, the phenotype is significantly more pronounced (new Figure 5a, 5b, and 5e).
- The reason the dynamics in the old Figure 5D, E looks the same is because we show it for a cell that did not change cell fate.

Second, in Fig. S1C, Hes1 dynamics seems to be almost independent of Dll1 (but they do depend on Jag1 in Fig. S1D).

- Indeed, comparing old Figure S1B and S1C (e.g. middle panels) does not show much difference between Wildtype and Dll1-deficient cases. As we mentioned above this situation corresponds to the case when the cells didn't change their cell fate. However, looking at the bottom panels in old Figure S1B and S1C, it is clear that for they change cell fates in Dll1-deficient case (from two BP cells to one BP and one PAC). One should however remember that these are results of a 2-cell model, and the specific cells fates will be different in a 3D model as difference in relative magnitudes of *cis*- vs. *trans*-interactions arise from different number of neighbors in 3D case.

It seems that the parameters controlling Dll1 (a_D and a_W) seem to be too weak compared to the parameters regulating the Jag1 to actually make a difference. If that is the case, then the fact that model 'predicts' different behaviors for Dll1 and Jag1 mutants is rather trivial – Dll1 doesn't really do anything. This contradicts Fig. 6G claiming Dll1-Hes1 feedback is important for MPC.

- We hope we have addressed the reviewer's concern of *Dll1 deficient* phenotype in our replies to points above.
- We however did appreciate the reviewer's point that the selected values of parameters might lead to bias in our conclusions. In the revised manuscript, we adjust the mentioned parameters and provide parameter sensitivity analysis to address this concern.
 - In the revised multi-cell model, we obtain similar results, with the similar parameters for regulation of Dll1 and Jag1 by Ptf1a. In the Hill function, we use the same Hill constant ($K_4 = K_6 = 4.0$) for Dll1 and Jag1, and maximum production rate regulated by Ptf1a of Dll1 is $a_w = 0.8$, of Jag1 is $a_J = 1.0$ respectively (see Methods).
 - We perform sensitivity analysis of all the parameters with the two-cell model (new Supplementary Fig. 7). In the two-cell model, $a_w = a_J = 1.0$, and the cell fate segregation is robust to the values of a_w and K_6 (new Supplementary Fig. 7b).

The main conclusions are conserved with these modifications. Thus, our new results support the idea that the difference of Jag1 and Dll1 dynamics is a result of the different regulations within the GRN.

As far as I can tell, in ref 14 it is shown that modulating the transcriptional delay of Dll1 generate observable phenotypes (affect Hes oscillations), which means that Dll1 is not obsolete in this stage. The authors should try to modify their model so that the Dll1 branch is relevant, and hopefully reproduce the effects observed experimentally. If the authors think that indeed the Dll1 branch does not do much, it should be omitted from the model altogether.

As mentioned above, the Dll1 branch is relevant both in the old and the modified version of the model. To validate our model further, we have tested if we can reproduce the result of the transcriptional delay on Hes1 oscillation as reported in (Shimojo et al., 2016). We find that:

- 1) The two-cell model for the pancreatic system recapitulated the “oscillation death” caused by delay in Dll1 transcription. In the neural progenitors, the Notch signaling is limited to Dll1, which is close to the scenario of removing Jag1 in our two-cell model. In line with results in (Shimojo et al., 2016), when we embedded a time delay in the production terms for Dll1, we found the “oscillation death”. This was observed when the delay was in a narrow time window around 10 min (new Supplementary Fig. 2g–h). The oscillations recovered but the frequency is changed if the delay was prolonged to 18 min. These results are consistent with the previous mathematical model which represented the cell’s state with one variable (Shimojo et al., 2016). Importantly, these results validate that the response of Dll1 in the circuit simulated with our model is critical for observed Hes1 oscillation with the selected values of parameters.
- 2) In a multi-cell model for pancreatic development there is no oscillation death. The 6 min delay in Dll1 transcription only reduced Hes1 amplitude (in absence of Jag1) (new Supplementary Fig. 4d). This is most likely because each inner cell has at least 9 neighbors and there will always be some Notch signaling to activate HES1. However, the delay in Dll1 changed proportions of cell fates (new Supplementary Fig. 4i). The longer the delay is, the fewer MPCs the E12.5 pancreas has. The proportion of MPC cells decreased to 3% from 15% when the delay was 15 min, indicating that prompt response of Dll1 to Hes1 is critical for MPC state maintenance in pancreatic development.
- 3) Interestingly, this result is in good agreement with the in vivo observations. Both simulations and experimental results show that compared to wildtype, deficiency in Dll1 leads to reduction in MPC proportion at the final stage (new Figure 5a, Supplementary Fig. 4a). In the simulation, the Dll1 deficient mutant (Dll1^{ΔF_{oxa2}}) does not have MPCs at E12.5 (Figure 5a, Supplementary Fig. 4a). (In the original manuscript, the trend was the same, however there is only one MPC in the wild-type simulation at the final stage and no MPC in the Dll1 deficient simulation)
 - a. The model also predicts the specific roles for *cis*- and *trans*-interaction of Dll1 at the early time of differentiation. Without *cis*-interaction of Dll1, the simulation shows that the cells are decoupled from each other as high levels of Dll1 lead to higher Hes1 (and consequently no Jag1). These cells can not undergo cell fate segregation (new Supplementary Fig. 5c). Without *trans*-interaction of Dll1, the duration of MPC state at early times become shorter (Supplementary Fig. 5d), indicating Dll1 supports transient MPC state through *trans*-interaction.
- 4) The role of Dll1 in MPC maintenance could explain the phenotypes in ref 14: Pancreatic bud size has comparable reduction in type1/type2 Dll1 mutants and Dll1^{ΔF_{oxa2}} embryos. While it increased in Jag1^{ΔF_{oxa2}} embryos. Delayed or

depleted Dll1 expression hampers MPC expansion thus lead to smaller pancreas size.

Taken together, in our model Dll1 is relevant to the cell fate segregation process, and we conclude that Dll1 contributes to MPC state maintenance and other biological observations.

2. I think there are a few technical problems with the definition of the model and its implementation that need to be clarified:

2a Implementation of Dll1 deletion. To implement the deletion the authors set a_D to 0 (line 381, methods). I think they need to set both a_D and a_W to 0, since in the deletion there should be no expression of Dll1 at all.

We are sorry for this mistake, in the revised version, we set both a_D and a_w to 0. This adjustment did not change the conclusions.

2b In line 370 the authors use unusual normalization for interaction term (e.g. $[ND_t] = (ND_t)/(N+D_t)$). It should be explained why this normalization is used for a biochemical reaction. Why not use standard biochemical equilibrium expression $[ND_t] = (ND_t)/k_t$ as is typically being used?

We thank reviewer for reminding us of the missing explanation. We explained these terms more clearly in the manuscript now. The reason we use $[ND_t] = (ND_t)/(N+D_t)$ the term and not $[ND_t] = (ND_t)/k_t$ is that the latter does not capture the case when e.g. ligand at low concentration will be saturated by receptors and increasing amounts of receptor in this case should not lead to increased signaling. The expression can be derived from mass conservation and assumption of tight binding between the receptor and ligand.

Within only the process of binding, if current concentrations of notch receptor and Dll1 ligand are N and D , the concentration of complex is $[ND]$, we have $N = N^{total} = N^{free} + [ND]$ and $D = D^{total} = D^{free} + [ND]$. On the other hand, at the steady state, $[ND] = N^{free}D^{free}/K_D$, K_D is the dissociation constant of the binding.

Such that, $[ND] = (N^{total} - [ND])(D^{total} - [ND])/K_D$.

With $[ND] \ll \min\{N^{total}, D^{total}\}$, the term $[ND]^2$ could be omitted,

$[ND] = (N - [ND])(D - [ND])/K_D \rightarrow [ND] \approx ND/(K_D + N + D)$.

$K_D \approx 0$ for tight binding, and the binding process is fast compared with the gene expression. When wired in the equations for gene regulations, the *cis*-interaction could be approximately represented with an additional limited rate γ_1 as $\gamma_1[ND] = \gamma_1 ND/(N + D)$.

This mathematical approximate method is used for all *cis*-interactions and *trans*-interactions between Notch receptor and Dll1/Jag1 ligands.

2c In line 375 (model equations) – In the Delta equation, why are the activation by P and repression by H are additive and not multiplicative (as in the equation on H). This creates a situation where even at very high H levels, there is no complete suppression. Please justify this assumption.

The reviewer mentioned a very important point about the rationality the assumption for the model. In the original manuscript, we used the additive terms with the consideration of the biological fact that Ptf1a is absent in BP cells but Dll1 oscillates at a low level in these cells. This fact indicates Dll1 can be expressed without Ptf1a, which can not be recaptured by multiplicative terms.

We also realized that validating and clarifying the dependence of Ptf1a in regulation of Dll1 and Jag1 is very necessary and fundamental for the model. Thus, we conduct experiments with Ptf1a knockout mice to support the assumption. In the knockout mice, Jag1 is lost without Ptf1a expression. While Dll1 is reduced but not lost (Supplementary Fig. 1).

The experiments validate the assumption and provide evidence for that the additive terms should be used when integrated the input from Hes1 and Ptf1a to Dll1.

3. Another main point in the manuscript is that Hes1 indirectly regulates Jag1 through Ptf1 (compared to direct regulation of Dll1). According to the manuscript this is important since Ptf1 has a slow decay rates and serves as a time average of Hes1 expression. To really show this point the authors need to compare 3 models: (i) The model they suggest. (ii) A model with fast Ptf1 decay rate. (iii) A model where Hes1 directly regulates Jag1 (as it does for Dll1). Comparing these models will help the authors support their claim regarding the role of Ptf1 as a buffering and averaging element.

We thank the reviewer for this suggestion. In the revised manuscript, we add some new results about how time scale of Ptf1a affects cell fate segregation. We applied and compared different time scales of Ptf1a in the equation: slower (0.5 time of wildtype), faster (2 times of wildtype) and extremely fast (50 times of wildtype). The simulations show that the time scale of Ptf1a determines the timing of cell fate segregation (Supplementary Fig. 6a–6c). Faster Ptf1a results in faster PAC and BP differentiation. When Ptf1a response to Hes1 extremely fast, which is equivalent to have Hes1 directly regulate Jag1, the cell fate segregation happens immediately (Supplementary Fig. 6d–6f). As expected, Jag1 shows oscillatory dynamic similar with Dll1 in MPC state.

These results suggest the role of Ptf1a as a buffering and averaging element in the gene regulatory network, and we discuss these results in the revised manuscript (Line 355–362).

4. The claims in Fig. 6 and Fig. S4 are unclear. First, Figs. 6B-C only show what happens in WT with high cis-inhibition but not with low cis-inhibition.

We are sorry for the confusion. We are not sure why this figure was unclear, as in Figure 6B–C the x-axis represents the entire range of cis-inhibition from low to high. This figure has been now modified to better reflect the proportion of cells for each value of *cis*-interaction strength (new Figure 6a, 6b). We run the simulation with each value of γ_1 on the x-axis of the heatmap and show the distribution of amplitudes of Hes1 in all of the cells at the final stage. Each column of the heatmap corresponds to a simulated pancreas (new Figure 6a) (or without Jag1, new Figure 6b).

It is unclear what Fig. S4 is trying to show.

This figure was showing the results in absence of *cis*-interactions but different strength of trans interactions. We have now removed this figure as it similar results are now discussed in new Figure 2, new Supplementary Fig. 2 and 7.

The authors should show Figs. 6B-C without cis-inhibition (or how it changes with cis-inhibition strength).

Fig. 6E-F show that in Jag1 deletion, there is no real lateral inhibition. This is not surprising given the issue raised in point 1 above (since it's the main branch through which interactions occur).

We hope we have addressed the concerns raised in Point 1 above. Also, our updated model where Ptf1a regulations on Dll1 and Jag1 are now comparable (new Figure 6b), produces more intuitive outcomes, where one can see that lateral inhibition from Dll1 can still lead to some cell fate segregation, albeit requiring significantly stronger cis-interaction rate (γ_1)

5. The authors assume the same strength of cis-interaction for Dll1 and Jag1. This is likely not the case as it has been shown that strength of cis interactions may be very different between Dll and Jag and may be modulated by fringe (see for example LeBon et al eLife 2014). It could be that the different phenotypes associated with Dll1 and Jag1 may be related to this property and not to the difference in GRN. Can the authors test that option (Or at least argue whether this should be important or not)?

We thank the reviewer for pointing us to this reference. The results in LeBon et al show that the Dll1 has a greater affinity than Jag1 in the binding activity with Notch receptor. We thus test our model with a smaller *cis*-interaction rate for Jag1 (γ_1 for Dll1 is 0.25, and for Jag1 is 0.15). Interestingly, this alternation of parameters does not change the simulated phenotypes of Dll1 or Jag1 deficient (compare new FigurS4A-D with the Response Figure 1) indicating robustness of these parameters in case of different regulations on Dll1 and Jag1.

Response Figure 1. Simulations of Wildtype, Dll1 deficient mutant, and Jag1 deficient mutant with different *cis*-interaction rates (γ_1) for Dll1 and Jag1.

Furthermore, in addition to our new experimental results supporting the differences in Dll1 and Jag1 regulations, we have also tested computationally the case where both Jag1 and Dll1 are regulated directly by Hes1 and find that the differences (direct regulation of Dll1 by Hes1 and indirect regulation of Jag1 through slow Ptf1a) are essential.

For this, we considered a model where Ptf1a is short-lived, which is equivalent to that Hes1 regulates Jag1 and Dll1 directly (new Supplementary Fig. 6d–6f). We can see that this assumption leads to 1) Immediate cell fate segregation, thus lacking the transient MPC population observed in vivo (Seymour et al 2020). 2) oscillatory dynamic of Jag1 similar to Dll1 at MPC state which contradicts the uniform distribution of Jag 1 in vivo (Seymour et al 2020, see also our arguments for defining MPC state Figure 2A).

6. Parameter choice – The authors do not explain how they chose the parameters for the “WT” model. I think they should add a short explanation that answers the following questions: Are the chosen parameter values based on experiments? Are they fine tuned to match experimental results? If so, how sensitive are the results to a small change in each parameter?

We apologize for omitting this important information. Some of the parameters are derived from experiments and the others are fine-tuned to match the experimental results. We are now listing the source of the parameters in Methods, Table1.

Per reviewer’s suggestion we performed sensitivity analyses of all the parameters in the model by changing their values from 1% to 200% of the values chosen for wildtype. The analysis shows that the model is robust to 50% increase/decrease in most of the parameters (Supplementary Fig. S7–S8). Interestingly, this analysis, shows that some parameters (e.g. a_J and τ_p) are able to cause bifurcation, resulting in e.g. two possible Hes1 amplitudes/periods at a particular value of the parameter.

7. The time delay for Hes1 chosen in the model is 40 mins. The oscillations in the model (for example Fig. 2) seem to have a period close to the observed 90 mins. Is the difference coming from the coupling to Notch? How does the oscillation period depends on the time delay?

This is a very good question. We have extensively explored which parameters could change the period of Hes1 (new Supplementary Fig. S8) and found that in total 9 parameters have an impact (in (at least one parameter in each of the equations). More interestingly, while the period increases continuously with time delay (τ_0), period changes in a stepwise manner with changes in other parameters. In addition to the above, the delay in Dll1 transcription also changes the frequency of oscillation.

8. For the 3D model, the authors need to show statistics indicating if the results are simply because of edge cells having less inhibition from their neighbors? This can be tested by considering geometry with periodic boundary conditions. Is Hes1 in the actual system show lower expression at boundary cells?

We thank the reviewer for the inspiring advice.

We have now added statistics on the number of neighbors as well as position from the pancreas center (new Supplementary Figure 3d and 3e).

To clarify the boundary effects in cell fate segregation, we simulated a model where each of 143 inner cells had the same (12) number of neighbors. In this case PAC cells are not preferentially located at the surface anymore (Supplementary Fig. 3f–3h). Thus, in wildtype, PAC cells prefer to be on the surface since they have less inhibition to Ptf1a from Hes1. In vivo, Hes1 is low in PACs, which are mostly located at the boundary (Figure 1 and Seymour et al, 2020).

Detailed list of minor issues:

9. In the abstract, line 10, it seems more appropriate to replace “The model predict...” with “the model captures”

We changed the word “predict” to “capture” in the revised abstract.

10. In the abstract, line 15-16 the claim is that trans-activation feedback is associated with Dll1, but this is not manifested in the model. (see major point 1)

We hope our explanations in this reply and modifications to the model and the manuscript are now sufficient to keep this claim as is.

11. On line 38 the authors claim that “ligand degradation is facilitated by cis-interactions”. Is there an experimental evidence supporting this claim?

Sorry for confusion, we did have a reference included, but it was not inserted correctly. We now modified this sentence. After the *cis*-interaction, the complex of interacting receptor and ligand is transported to lysosome for further degradation (K. G. Guruharsha, et al. 2012). In the model, we assume that these receptors and ligands are not considered in the modeling process anymore after the *cis*-binding.

We have now added this and a few other references in the right place in the text.

12. In line 113 the wrong figure is referenced, should be “Figure 1F”.

Thanks, we corrected it in the revision.

13. In line 118-119 the explanation about the different simulations should be more accurate. There is no simulation with strong trans-interactions and weak cis-interactions, those are 2 simulations, one with strong trans-interactions and medium cis-interactions and one with medium trans-interactions and weak cis-interactions (at least this is what is shown in figure 2).

Thanks, we have now carefully revised the explanation.

14. In line 126 a discussion about what does “strong” or “weak” cis- or trans-interactions mean in terms of model parameters is required. Is there some criteria describing what strong and weak means?

We meant compared to wildtype. We have now modified the sentence as:

“With strong *trans*-interaction (K_2 is 0.15 compared with 0.5 for wildtype) or weak *cis*-interactions (γ_1 is 0.1 compared with 0.25 for wildtype), Hes1 exhibits comparable anti-phase oscillations in two interacting cells.”

And “Finally, when the *trans*-interaction is weak (K_2 is 0.9 compared with 0.5 for wildtype) or *cis*-interaction is strong (γ_1 is 0.4 compared with 0.25 for wildtype), the cells exit the MPC fate rapidly and directly bifurcate into the respective downstream cell fates”.

Hope the modified sentences are more precise.

15. In line 128 the word “which” should be deleted (typo).

Thanks, we deleted it.

16. In line 150 it should be clearly stated that it is about the 2-cells model.

We modified this sentence as suggested.

17. In line 158 the authors say that in the simulation results PACs have high Ptf1a expression and low Hes1 expression, compared to BPs. This should be quantified.

The sentence is modified now as “The PACs have high Ptf1a expression (the average is 9.4 and 47.2 times higher than in MPCs and BPs, respectively) as its repressor Hes1 is low...”

18. In line 163 a comparison between the number of neighbors of each cell in the simulation and the real tissue should be added. Can the number of neighbors reach 14 (as in the simulation)?

In the revised manuscript we now compare the number of neighbors in our 3D structure with the number of neighbours seen in vivo, estimated from the number of neighbours seen in a single plane in confocal microscopy. (Line 231–236)

19. In line 192 authors say that changing K_2 corresponds to DAPT and MLN4924, explain why this is assumed.

We have now added an explanation in the Methods and Manuscript.

“... $K_2 \propto 1/K_c$...The small-molecule treatments perturbed NICD levels in cells by inhibiting NICD production (DAPT) or inhibiting NICD degradation (MLN4924),

indicating that K_c is smaller or larger with the same level intercellular interaction of receptor and ligand (γ_2). Thus, K_2 was increased to simulate the effect of DAPT treatment and was decreased to simulate the effect of MLN4924 treatment.”

20. In line 225 explain how can one see, by examining the model results, that “DII1-mediated trans-activation helps maintain the MPC fate when Jag1 expression is low at the early stage”. (This is related to major point 1 above)

We hope our edits and reply to Point1 above resolves this concern.

21. In line 360, explain how a neighbor is defined in the simulation.

In the revised Methods, we now explain how we define the neighbors of cells (Line 553–556). “Regarding the cells’ current locations, the cell_j was defined as a neighbor of cell_i if cell_j had shortest distance to the midpoint of these two cells compared with any other cells. If cell_j was a neighbor, it was assumed to have direct contact with cell_i.”

22. In line 364 replace “Figure 1H” by “Figure 1F” (typo).

Thanks, we have corrected this typo.

23. In line 375 (model equations) – A different notation is used for the same thing in different equations, sometimes using the $[\cdot]$ notation for normalized terms and sometimes writing them explicitly.

Thanks for pointing this out to us, we have now edited this part and use $[\cdot]$ in all equations.

24. In table1 the description of the half time parameters is wrong (remove “Notch and” from all, looks like a typo).

Thanks for catching these typos, we have now corrected them.

25. In figure 3, sub-figure D bottom graph – It looks like the PAC result is missing.

Thanks, we modified the figures, now the results of PAC can be seen clearly.

26. In figure 4 mark K-M as experimental results in a clear way.

We have edited the figure to emphasize these results are from experiments and also mention it in the figure caption.

Reviewer #2 (Remarks to the Author):

The manuscript entitled “Jag1-Notch cis-interaction determines cell fate segregation in pancreatic development” by Xiaochan Xu et al. presents an elegant study of cell fate choice mediated by Notch signaling in the pancreas. The study focuses on the choice from multipotent progenitor cells (MPC) to pro-acinar cells (PAC) and duct/endocrine progenitor cells (BP). The results mostly come from modeling (simulation results). The study is based on experimental data previously published by the laboratory of Pallo Serup, who is author of the manuscript reviewed here, and provides few new data.

The results present a model for Notch signaling arising from ligands Dll1 and Jag1. The simulation results recapitulate the in vivo expressions and suggest that cis-inhibition mediated by Jag1 is crucial for cell fate choice. The authors set into the model that Hes1 oscillates through a delayed negative feedback, as previously suggested, and drives oscillations of Dll1 by repressing it. Jag1 and Dll1 trans-activate and cis-inhibit Notch signaling. Jag1 does not oscillate because it is regulated by Ptf1a, which has a slow decay and therefore is a slowly evolving variable that can average Hes1 oscillations. The formulated model seems correct based on the hypotheses and knowledge so far and the obtained results seem reasonable.

In my opinion the manuscript provides a plausible model for this developmental process. It represents an effort to formulate a framework capable of reproducing the observations. And this effort is achieved. The model reproduces the expressions of the selected genes in the wild type and the change in proportions of cell fates when Notch signaling is altered. Taken together, the study shows a plausible model for how MPC to PAC/BP fate choice occurs. From this plausible model, the authors extract new knowledge: that Jag1 cis-inhibition is crucial.

However, there is no data to validate the model and this prediction. The same data that are used to construct the model are the ones that the model reproduces (i.e. expression data in the wild type and changes upon Notch alteration, which sustain that Notch and lateral inhibition is relevant). In my opinion, validation of the model requires: A) to test whether additional factors that need to be assumed in the model are really happening in vivo or not (or to prove this is the only framework that works (what is an impossible task)) and B) to test its predictions (i.e the crucial role of Jag1 cis-inhibition). Thus, at present, the model is plausible but is not tested. In this sense, I believe that the manuscript does not provide enough new findings, since it does not validate its proposals, or some of them.

We apologize for not being clearer in describing how the model was constructed. The model was constructed solely from expression patterns in wildtype, published ChIP-seq data for molecular links between the transcription factors (TFs) and the ligand genes and, in this revised manuscript, also from changes in ligand expression upon homo- and heterozygous deletion of Ptf1a (new Figures 1f and S1a), again to strengthen the data indicating molecular links between TFs and ligand genes.

Importantly, cell fate changes upon Notch alteration in vivo, whether genetic or small molecule based, was not used to construct the model. Instead we tested the ability of

the model to capture the changes observed in vivo, when recapitulating Notch alterations in silico. In this revised manuscript we additionally test the ability of the model to accurately capture the in vivo effect of changed transcriptional delay of Dll1.

Here below I indicate my specific major concerns:

1) In the model, there is a clear asymmetry between Jag1 and Dll1. Jag1 is controlled by Ptf1a while Dll1 is mainly controlled by Hes1. Because Ptf1a is a slowly evolving variable, Jag1 does not oscillate, whereas Dll1 does. This asymmetry explains why simulations that are deficient in Jag1 drives distinct results than those that are deficient in Dll1. But:

1.a. What are the experimental evidences for the slow decay of Ptf1a? This slow decay is crucial to obtain the results presented in the manuscript. To validate the model it is necessary (among other things) to provide evidence for this slow decay.

We agree with the reviewer that the slow decay of Ptf1a is crucial for the results presented and apologize for not clearly pointing out the evidence for slow Ptf1a decay. We have now remedied this point by both providing the references for a slow decay as well as showing it experimentally by measuring decay in 266-6 cells after cycloheximide treatment.

The half-life of Ptf1a has been reported to be ~2.5 h in HEK-293T cells and ~3.5 h in 266-6 cells, indicating a slow decay (Hanoun et al. (2014) *Journal of Biological Chemistry* 289: 35593-604). Our measurements in 266-6 cells showed a half-life of ~2.5 h (new Figure S1c), consistent with the published data.

Similarly, Jag1 has been reported to have a half-life of ~4 h in HUVEC cells (Dos Santos et al. (2017) *Oncotarget* 8: 49484-501), a slow decay compared to Dll1's reported half-life of ~50 min (Shimojo et al. (2016) *Genes Dev* 30: 102-16). Our measurements in 266-6 cells showed a Jag1 half-life of ~3.5 h (new Supplementary Fig. 1d), consistent with the published data.

1.b. The model sets that Ptf1a regulates weakly Dll1 while it strongly regulates Jag1: what is the evidence in favour of this difference? Supporting this would also provide a partial validation of the model.

This is an excellent question and we thank the referee for raising this point. To our knowledge there is no published evidence favoring this notion. To begin to address this question we generated E12.5 homo- and heterozygous Ptf1a mutant embryos and analyzed Jag1 and Dll1 expression compared to wild types. As shown in the new Figure 1f and Supplementary Fig. 1a, Jag1 expression is reduced in the pancreatic epithelium of Ptf1a^{+/-} embryos and lost in Ptf1a^{-/-} embryos, while Dll1 expression was unchanged and reduced, respectively, in these embryos. While not formally demonstrating that Ptf1a is a stronger activator of Jag1 than of Dll1, these results do show that the expression of Jag1 is more dependent on Ptf1a than expression of Dll1 is, which is consistent with the notion of Ptf1a being a stronger activator of Jag1 than Dll1.

(In addition, references that support that Hes1 represses Ptf1a should be provided.)

We apologize for the lack of clarity. we did cite evidence for Ptf1a being inhibited by Hes1 in the introduction (line 35). The references are: de Lichtenberg et al. (2018) BioRxiv 336305, Fukuda et al. (2006) J Clin Invest 116: 1484-93, and we have now revised the corresponding sentence for better clarity.

2) In the model there is another asymmetry: γ_1 , which is the rate of (degradation by) *cis*-interaction, is much larger than γ_2 , which is the rate of (degradation by) *trans*-interactions.

2.a. What is the evidence for this strong difference? Assessing this will provide a partial validation of the model.

We agree with the reviewer that the big difference between γ_1 and γ_2 should be carefully considered. Unfortunately, we have not found the direct experimental evidence showing that the binding activity of ligand and receptor in *cis*-interaction is faster than *trans*-interaction. However we addressed this point computationally in several ways:

- 1) One of the reasons γ_2 was smaller than γ_1 was because when calculating *cis*-interactions, it was only for one cell, whereas in calculating *trans*-interactions we were summing effects from all the neighbors and γ_2 had to be smaller than γ_1 to account for multiple interactions. We have now modified the model by replacing the sum with the average receptor/ligand interactions from neighbors. This allowed to bring the two parameters closer together, but the asymmetry remained to be important.
- 2) By exploring phenotypes at different combinations of γ_1 and γ_2 , we find that
 - a. In the two-cell model we performed sensitivity analysis of γ_1 and γ_2 (new Supplementary Fig. 7c) and find that the cell fate segregation improves (i.e. the separation between Ptf1a levels is increasing) with increasing γ_1 and decreasing γ_2 . This indicates that these terms have opposing effects. We can bring $\gamma_1 = 0.25$, $\gamma_2 = 0.1$ (new Figure 2). However, the two-cell model is less constrained as it can only resolve up to two coexisting cell states.
 - b. In the multi-cell model, where 3 cell states co-exist, a smaller $\gamma_2 = 0.02$ is needed to recapitulate experimental observations in *Dll1 deficient* mutant. There, larger γ_2 results in BP cells with very low Hes1 amplitude, making it hard to distinguish BP cells from MPC.

Overall, our exploration of these parameters suggests that strong *cis*-interaction removal of then [N:D] complex amplifies the difference between N and D inside the cell, thereby generating mutually exclusive GRN states as also earlier suggested by Printz et al. (see also Figure S7, γ_1 plot). On the other hand, strong *Trans*-removal of [N:D] reduces the difference between neighboring cells, e.g. if cell 1 has high N and its neighbor cell 2 has high D (or J), these will be rapidly reduced if γ_2

is high. In addition, low trans-[N:D] will only weakly activate Hes1, lowering the Hes1 level in BPs and thus the BP phenotype very close to that of MPCs (see point b. above).

2.b. In addition, while γ_1 is considered to be the rate or strength of cis-interactions (I agree with that) and it is changed to evaluate the effect of cis-inhibition, γ_2 is not changed when evaluating the effect of trans-interactions. Instead, K_2 or a_H are modified to evaluate the effect trans-interactions. In my opinion, a change in trans-interactions should involve a change in two parameters: in γ_2 and in a_H (or K_2). The authors should repeat their analysis of trans-interactions and cis-inhibition, by changing both γ_2 (and a_H or K_2) and γ_1 . This is relevant since the results claim that without cis-interactions (i.e. $\gamma_1=0$), despite changing trans-interactions (through K_2), there is no cell fate choice of BP and PAC. However, without cis-interactions, what happens if γ_2 parameter is as large as γ_1 default value?

We agree with reviewer that the strength of *trans*-interaction is affected by two processes in our model: 1) binding of ligand and receptor (governed by γ_2) 2) activation of Hes1 (governed by K_2). Specifically, for the bound notch receptor and ligand complex within *trans*-interaction, the outer membrane parts are also removed through endocytosis after the inner membrane part is cleaved. Thus, we tend to use the γ_2 as parameters only relevant to the process for binding. The outcome of *trans*-interaction is mediated by the cleaved inner membrane part of Notch receptor, which is calibrated by the parameter K_2 (the ratio K_2 : γ_2 matters) (see Methods). K_2 decides how strong the input to Hes1 transcription is by the activated Notch signaling (NICD).

In the original manuscript, tuning a_H to simulate DAPT and MLN4924 is improper. The a_H represents maximum production rate of Hes1 and does not change with DAPT and MLN4924 treatments. We removed this result in the revised manuscript.

In the two-cell model, we only discussed about the results by changing K_2 , which corresponding to results related to DAPT and MLN4924 in the 3D multi-cell model. These is because:

The DAPT is γ -Secretase Inhibitors, which inhibits the cleavage of the inner membrane part of Notch receptor (reduces NICD production). The MLN4924 is Nedd8-activating enzyme inhibitor, which inhibits the ubiquitinating degradation of NICD. These two conditions affect the processes after the intercellular interaction of receptor and ligand. They do not change the γ_2 . Instead, they should change K_2 , which quantifies the consequence from intercellular binding to Hes1 activation.

While as the reviewer suggested, analysis of how γ_2 can affect the cell fate segregation is interesting. We have performed sensitivity analyses of the parameter γ_2 : 1) Changing γ_2 alone, 2) Changing γ_2 and K_2 simultaneously.

From the first aspect, when γ_2 gets larger, even though it helps increase the *trans*-activation effect is also increasing the rate of removal of both receptor and ligand (*trans*-inhibition). If the removal has a dominant effect, the interacting neighboring cells tend to behave similar without any cell fate segregation. The simulation result of two-cell model is presented in Supplementary Fig. 7c (Response Figure 2a). The

result with 3D model is similar to the result with two-cell mode: The cells do not differentiate when γ_2 increases (Response Figure 2b).

From the second aspect, when changing γ_2 and K_2 simultaneously, with *cis*-interaction ($\gamma_1 = 0.25$), increasing γ_2 hampers cell fate segregation (Response Figure 2c). While without *cis*-interaction ($\gamma_1 = 0.0$), cell fate segregation fails regardless of K_2 and γ_2 (Response Figure 2d and 2e).

Response Figure 2. Increased parameter γ_2 hampers cell fate segregation. **a** and **b**, with *cis*-interaction, increasing γ_2 alone leads to cell fate segregation failure in both two-cell model and 3D model. **c**, with *cis*-interaction, increasing γ_2 leads to cell fate segregation failure with different K_2 . **d** and **e**, without *cis*-interaction ($\gamma_1 = 0.0$), cell fate segregation always fails regardless of K_2 and γ_2 . * data is also shown in manuscript.

2.c. A relevant validation would be to test whether cis-inhibition of Jag1 is indeed relevant or not. I understand that experimentally eliminating cis-inhibition is out of the scope. But what would be expected in Jag1 gain-of-function scenarios in the model? And in Dll1 gain-of-function scenarios? In these gain-of-function scenarios the effect of cis-inhibition and of trans-interactions could be tested in the model, and see whether they raise distinct predictions. Ultimately, this could be validated with data in gain of function embryos.

We thank the reviewer for the suggestion to extend the testing of our model.

We have simulated the Jag1 and Dll1 gain-of-function by increasing maximal production rate (a_J for Jag1 and a_D and a_W for Dll1, Response Figure 3) and find that Jag1 and Dll1 gain of function mutants have contrary effects in cell fate segregation. These together with the results of Jag1 and Dll1 deficient simulation (Figure 5a) show that increasing Jag1 facilitates MPC differentiation into PACs and BPs, while increasing Dll1 hampers MPC differentiation. Moreover, increased Jag1, when cis-interaction is simultaneously reduced, rescues the defective cell fate segregation seen with low *cis*-interaction rate (new Figure 6c).

We also investigate the contributions of *cis*- and *trans*- interaction of either Jag1 or Dll1 by removing each of the respective terms one at a time and monitoring component expression levels and cell fate segregation at early and late time points (new Supplementary Fig. 5). Removing *cis* interaction of Jag1 dramatically prevent MPCs differentiation at late time.

Response Figure 3. Jag1 gain-of-function increases MPC proportion, and Dll1 gain-of-function increases MPC proportion. In the manuscript, the MPC proportion of “Wildtype” is ~15%, and $a_J = 1.0$, $a_D = 0.5$, $a_w = 0.8$

3) The expression of MPC, PAC and BP cells needs to be clarified:

3.a. At stage 10.5 cells MPC cells have Ptf1a and Hes1 oscillates (line 90 and Figure 1A-B,D). They also have Jag1 (Figure 1A). However, in the simulations, MPC cells do not have Ptf1a nor Jag1 while they have Hes1 oscillating. How are these differences reconciled?

We are sorry for the not making this point clearer. In the simulations Ptf1a and Jag1 are expressed in MPCs, at low but significant levels (~1uM for Ptf1a and ~0.1uM for Jag1 in the final state, Figure 3c and 3d). They appear to be missing because we set

the scale to include other states with much higher levels of these factors. We have now added a sentence to the figure caption (new Figure 3) to make this point clear.

3.b. The model relies on the assumptions that MPC involve oscillations of Hes1 and Dll1 whereas PAC and BP fates correspond to sustained and distinct expressions: PAC cells express Dll1, Jag1 and Ptf1a, whereas Hes1 is expressed in BP cells. In Figure 1C, while I see that Hes1 and Ptf1a are expressed commonly in distinct cells and Jag1 co-localizes with Ptf1a, I do not see more co-localization of Dll1 with Ptf1a than with Hes1, or at least it is not obvious to me. Therefore, I do not see clearly that PAC cells have high Dll1 whereas BP cells do not. Could this be made clearer or clarified?

We agree that it can be difficult to see that PAC cells have high Dll1 whereas BP cells do not. One reason is that Dll1^{Hi} cells are not restricted to PACs when considering the entire organ. However, our description of the expression patterns was far from clear and we apologize for that. We have now revised the figure legend so that it now clearly explains what cell state the different arrows and arrowheads are indicating. This includes that peripheral Dll1^{Hi} cells are typically Hes1^{Lo}Ptf1a⁺Nkx6-1⁻, while peripheral Dll1^{Lo} cells are typically Hes1^{Hi}Ptf1a⁻Nkx6-1⁺.

Importantly, more centrally located Dll1^{Hi} cells are typically Hes1^{Lo}Ptf1a⁻Nkx6-1⁺, thus expressing the BP marker Nkx6-1. However, it should be remembered that Ngn3 is also a direct activator of *Dll1* expression (Ahnfelt-Rønne et al. (2012) Development 139: 33-45, Schreiber et al. (2021) Mol Metab. 53: 101313) and Ngn3⁺ cells are also Dll1^{Hi} at E12.5 (Seymour et al. (2020) Dev Cell 52: 731-47). Given that Ngn3⁺ cells arise from Nkx6-1⁺ BPs and the marker combination used for Figure 3c, we cannot distinguish between Ngn3⁺ and Ngn3⁻ cells in the central domain, but we do expect a certain number of Dll1^{Hi} cells to be present there due the presence of Ngn3⁺ cells at this stage.

Nevertheless, one can also find centrally located Dll1^{Hi} cells that are Ngn3⁻ in the E12.5 pancreas. What cell state these cells represent is currently not fully understood, but it is possible that they may represent the earliest stages of endocrine differentiation. This notion is consistent with many of these cells being in a Sox9^{Lo} state, while Sox9^{Hi} BPs are generally Dll1^{Lo} (see for example Seymour et al. 2020, Fig S1C).

In response to Reviewer 3, point 2 we have now also added quantitative assessments of co-expression of Dll1, Jag1 and Hes1 with Ptf1a and Nkx6-1 in the revised manuscript. The cell counts at E12.5 reveal that ~30% of the Ptf1a⁺ cells are Dll1⁺, while ~18% of the Nkx6-1⁺ cells are Dll1⁺ (new Figure 1d). Furthermore, only ~20% of the Ptf1a⁺ cells are Hes1⁺, while ~80% of the Nkx6-1⁺ cells are Hes1⁺ (new Figure 1e). Note that for these cell counts we have not distinguished between Dll1/Hes1^{Hi} and Dll1/Hes1^{Lo} cells.

Lastly, a point of clarification: Hes1 still oscillates in BPs, in the model as well as in vivo (Seymour et al. (2020)). However, as Hes1 is downregulated in emerging PACs and Dll1 oscillations are driven by Hes1 oscillations, the model does not show continued Dll1 oscillations in PACs. Whether this is also the case in vivo is currently

unknown as the bioluminescence analysis of Dll1 oscillations in Seymour et al. (2020) did not go on for long enough a time to determine this.

4) The results show proximodistal (PD) patterning, however, this needs to be further investigated. The authors indicate that the position of cells (being at the surface or not of the epithelium) is a relevant cue for lateral inhibition since cells at the surface interact with less cells than those at the interior. This positional cue drives cells at the surface to preferentially become PAC cells, compared to cells at the interior, and this results in PD patterning. This seems reasonable and to be expected from lateral inhibition as the authors clearly explain. However, the simulations (e.g. Figure 3C) have few cells at the interior, such that most of them have an adjacent cell which is at the surface. Therefore, the PAC cells that arise in the simulation are all or almost all at the surface. Therefore, a clear PD patterning is found. However, if the simulations had many more cells at the interior, such that their neighboring cells are also all at the interior, then I expect PAC cells to arise at the interior as well, and not only at the surface. Thus, if many more cells are simulated, I expect that the bias of PAC cells being found at the surface will be much less relevant (because cells at the surface will tend to become PAC cells but some cells at the interior will also become PAC cells). In this situation, the PD patterning would be much less apparent, and it may be thought to be a weak cue to account for the in vivo PD patterning (which is much more strong: with a majority of the cells at the surface being PAC). Thus, the authors should justify the number of cells used at the simulations and should run their simulations with higher numbers since as evidenced in Figs.2C,4H, embryos have many more cells than those at the simulations.

We appreciate the reviewer for this insightful comment and the suggestions to improve our modelling.

In the revised model, we increased the number of cells from total 200 (43 inner cells) to 400 cells with 143 interior cells, which is comparable with the embryonic pancreas at E10.5. Consistent with the original results, the PAC cells still distribute mainly at the surface and are surrounded by BP cells in the case of larger system (new Figure 3b, new Supplementary Fig. 3d). In the wildtype simulation of the larger pancreas, we also observed more cells in the center maintaining MPC fate (15%), perfectly matching in vivo phenotype (new Figure 3d). The fact that MPCs localize at the center is somewhat different from the in vivo where MPC are spread throughout. We believe this discrepancy may be due to our current model does not include cell adhesion, and consequently, the cell sorting resulting from differential adhesion.

While the PD patterning of PACs is robust to the size of simulated pancreas, we do find breaking of this pattern (with PACs distributed at both the surface and interior) in some other cases. For instance, the Dll1 deficient mutant, where cells are either PACs or BPs, we can see a few PACs in the center (new Figure 5b). Interestingly, in vivo, the PACs also appear sparsely in the interior pancreas at E12.5 (Seymour et al. 2020).

Minor comments:

1) In Figure 2: the action of cis-inhibition at very early times seems to necessarily occur through Dll1 and not Jag1 since Jag1 is not expressed at early times (although

experiments do not clearly support that, e.g. Fig.1). Please discuss and clarify if cis-inhibition of Dll1 or Jag1 is relevant at this early times (that of Dll1 and that of Jag1 could be removed). Depicting Jag1, Dll1 and Ptf1a would also help to clarify.

We thank the reviewer for a great suggestion, but as mentioned above Jag1 is expressed at a low, but significant level at early stages. Nevertheless, new simulation results show both *cis*- and *trans*- interactions of Dll1 are important for maintaining MPC state at early times (new Supplementary Fig. 5c–5d). Removing *cis*-interaction of Jag1 has only a mild effect at the early times but hampers cell fate segregation at late times.

2) To model inhibition of Notch signaling pathway by Nedd8-activating enzyme inhibitor MLN4924: the authors could change τ_n , which is the parameter for Notch degradation without binding to the ligand. Please discuss.

In the model, we simulate the effect of MLN4924 with decreased K2 since MLN4924 inhibits activated Notch receptor (NICD) degradation, which only appears after interaction with ligand. τ_n is the parameter for a basal degradation rate of the full-length Notch receptor, which to our knowledge is not affected by MLN4924. Nevertheless, as suggested by the reviewer, we analyzed how the value of τ_n affects the cell fate segregation process.

When analyzed with two-cell model, a larger τ_n (slow degradation) does not change the cell fate segregation because the free Notch receptor is mainly removed by *cis*- and *trans*-interaction. In contrast, a very small τ_n (fast degradation) leads to low Hes1 since it hampers production of NICD (Supplementary Fig. 7d and Supplementary Fig. 8d). In the multi-cell model, the cell proportions do not change with τ_n (Supplementary Fig. 6g–6i), which is consistent with the two-cell model. Intriguingly, when τ_n is longer, the cell fate segregation slows down. Timing of cell differentiation is also discussed in the manuscript regarding Ptf1a, we integrated the results in along with the results of Ptf1a.

3) The sentence “ Taken together, the results generated by our models fill a gap in the discussion about how cell-intrinsic feedback can be crucial for cell fate choice, a feature that is absent in most of the theoretical models of Notch signaling. “ should be made more precise. The effect of cell-intrinsic feedback mediated by cis-inhibition can be seen in Ref [5], Formosa-Jordan et al. Plos one 9, e95744 (2014), Corson et al. Science 356, eaai7407 (2017), Bocci et al. Front Physiol 11:929 (2020), among others.

Thanks for reminding us. We added the references and modified this sentence on Page 18, Line 451–454.

4) In the model, while Jag1 deficient has no Jag1, Dll1 deficient has little Dll1 through a_w which is not set to zero. It would be better to have $a_w=0$ in this deficient scenario, or to justify otherwise.

We agree with the reviewer’s comment. We now simulate the Dll1 deficient mutant with both $a_D = 0$ and $a_w = 0$ in the revised manuscript (new Figure S2) and obtain the same results.

5) Panels D and H of Figure 2 are exactly the same, if I am not wrong. Please indicate so, or just keep only one of them.

We agree with the reviewer. In the revised Figure 2, we only keep Figure 2d.

6) I have not been able to find the files of Supplementary Movies 1-3.

Sorry for the inconvenience, we uploaded the movies in this revision instead of providing links.

7) What is meant by “sorted cells” in Figure S2E?

In the old figure, we sort the cells by the number of neighbors they have. We now replaced this figure with Supplementary Fig. 3e, where the numbers of epithelial neighbors, mesenchymal neighbors and total neighbors for each cell fate are shown in a better way. We thus no longer use the term “sorted cells”.

8) Line 169: Not clear the meaning that “there is an initial salt and pepper pattern that later develops to a proximodistal patterning. “ If I understood correctly, the simulations do not show re-arrangement of cell fates and hence there is not an initial pattern that after a while is re-organized proximally in a distinct manner. Please rewrite the sentence to clarify what the simulations show and what is thought to occur in the embryo.

We apologize for the confusing use of the term “salt-and-pepper pattern”. This sentence was meant to emphasize the initial alternating pattern of emerging PACs and BPs often seen in the periphery of the E12.5 pancreas (see for example Figure 1c). Although it is not a typical random “salt-and-pepper” pattern, it is noteworthy that this pattern of PACs and BPs are seen at the surface of the organ.

We have now modified the sentence to: The model thus presents a theoretical explanation of how spatial cues in the developing organ contribute to achieving the correct PD distribution of cell fates. (Line 241-242)

9) Why Sox^{Hi} and Jag⁺ notation and not simply “⁺” or “^{Hi}}” (not both notations)? In addition, in the abstract it is used “Jag^{Hi}PAC” and it remains unclear what is meant for.

The reasons we use this notation is that it is sometimes important to distinguish between “Hi” and “Lo” states, particularly for proteins that oscillate like Hes1 and Dll1. However, at other times, for example when we quantitate the total number of Hes1-expressing cells in the Ptf1a⁻ or Nkx6-1-expressing compartments it is most straightforward to refer to these compartments as Ptf1a⁺ and Nkx6-1⁺ and how big a fraction co-express Hes1, i.e. is Hes1⁺ or Hes1⁻, without distinguishing between Hes1 “Hi” and “Lo” states.

Even for other transcription factors it is sometimes relevant to distinguish between “Hi” and “Lo” states, for example, Sox9 is expressed at a low level in Ptf1a⁺ PACs

(but only revealed by some antibodies) but at high levels in bona fide $Nkx6-1^+$ BPs. However, even in the BP compartment one can find “ $Sox9^{Lo}$ ” cells, which are likely an indication of these cells being on the path to endocrine differentiation where $Sox9$ is downregulated early in the differentiation process such that distinct $Sox9^{Lo}Ngn3^{Lo}$ cells progress to become $Sox9^-Ngn3^{Hi}$. Lastly, the “mixed” $Hi/Lo/+/-$ notation is consistent with the notation used in Seymour et al. 2020 and has also been used by other authors, e.g. Bechard et al. 2016, *Genes Dev* 30:1852-65.

10) Figure 4H-J are from data in ref.[14]. Also Figure 5G-I. Please indicate.

We would like to clarify here that the image data in Figure 4H-J and 5G-I are from similar experiments as in Seymour et al. 2020, but the images shown in this paper are previously unpublished images from new stainings.

11) Figure S3: not clear to me that DAPT, when modeled as an increased K_2 , drives MPC fate or just a PAC with lower $Hes1$ amplitude (since $Ptf1$ is very low in these cells). Not clear also which is the proportion of PAC cells compared to DMSO. Please justify and clarify.

The DAPT drives MPC fate because both $Ptf1a$ and $Hes1$ are present at intermediate levels and thus are more similar to MPC state that to PAC in Wildtype (Response Figure 4 with Figure 4D)

We think this confusion may have arisen from the fact that $Ptf1a$ in PACs is very high with the DAPT treatment there by extending the range of x-axis in Figure 4d, thus the $Ptf1a$ in MPC appears very low. If we zoom in the left part of the figure (Response Figure 4), we can see that the MPCs with DAPT are very close with MPCs in DMSO condition. In addition, from the equation (5), we can infer that the final steady state of $Ptf1a$ depends on the level of $Hes1$. When the lower $Hes1$ is, the higher $Ptf1a$ is.

Response Figure 4. Zoom in plot of new Figure 4d.

We agree, it is a good idea to quantify the proportion of PAC cell proportions with DAPT. In the simulation, 24% of the cells become PAC with “DAPT”, which is a slightly higher proportion compared to “DMSO” (20%). We add this number to the figure caption. This change in PAC proportion is consistent with the experimental result showing that the DAPT increases the PAC from 38% to 57% in *in vivo* pancreas, although exact numbers do not match (Figure 4I).

12) Please correct errors in the definition of tau_n and tau_p in the Table of parameter values.

The typos are removed, and the terms are corrected in the revised manuscript.

13) Introduction: Lines 34-44: more references for the findings that are mentioned are needed. What reference supports that Hes1 represses Jag1 through Ptf1?

That Hes1 represses Jag1 through Ptf1 is inferred from the following published observations: HES1 binds to the *PTF1A* promoter in human ES cell-derived pancreatic progenitors (hESC-PPs) (de Lichtenberg et al. (2018) *BioRxiv* 336305) and Ptf1a is ectopically expressed in *Hes1*^{-/-} mutant mouse pancreas (Fukuda et al. (2006) *J Clin Invest* 116: 1484-93; Horn et al. (2012) *PNAS* 109: 7356-61).

Similarly, Ptf1a binds to the *Jag1* gene and *Jag1* is downregulated in *Ptf1a* mutant mouse pancreas (Meredith et al. (2013) *MCB* 33: 3166-79; this manuscript: new Figure 1f). Furthermore, PTF1A binds the *JAG1* gene in hESC-PPs and *JAG1* is downregulated in PTF1A-deficient hESC-PPs (Miguel-Escalada et al. (2022) *Dev Cell* 57: 1922-36).

14) Why data on oscillations of Hes1 are reanalyzed? What do we learn? Lines 98-106 state this re-analysis but it is unclear what it is useful for in this study.

We include the re-analyzed data to the manuscript to highlight the periodicity of Hes1 oscillations and for the convenience of readers.

15) Lines 89-90: add reference and cite Figure 1A.

We thank the reviewer for this advice, we modified the sentence by adding the reference and now cite Figure 1a.

Reviewer #3 (Remarks to the Author):

In this report by Xu et al authors examine the differentiation of pancreatic multipotent progenitors (MPCs) into pro-acinar cells (PACs) and bi-potential ducto-endocrine progenitors (BPs) using mathematical modeling that couples what is known about dynamic Notch signaling in these cells with the spatial distribution of interacting cells. The authors back up these predictions with experimentation, using both genetics and pharmacological approaches. This study tackles the challenging question of how the first major fate restriction happens in the pancreatic epithelium, that of acinar versus ducto-endocrine. The authors hypothesize that the salt-and-pepper distribution of progenitors could be governed by Notch-ligand mediated lateral inhibition. Furthermore, they incorporate cell-autonomous cis-inhibition and trans-activation mediated by cell-cell interactions in their model, which together predicts how cells with restricted potential (acinar vs ducto-endocrine) sort out into the correct spatial locations (acinar at the tips in the pancreatic periphery vs ducto-endocrine at the core of the pancreatic bud). An interesting prediction is based on the recent finding that downstream of Notch, Hes1 displays oscillatory expression in pancreatic progenitors and this the quality of this oscillation drives MPC and BP fate.

Change in oscillation frequency, modulated by slow responsive Pft1a leads to Jag1 activation and cell fate bifurcation. Similar fate restrictions based on analogous GRNs in the nervous system provides a road map for this study.

This study tackles a difficult question regarding cell fate within the early pancreatic epithelium. The work is based on observations in a paper by some of the authors in Dev Cell in 2020. The manuscript is dense, however, and difficult to get through, as the dynamic interactions are inherently complex. But the methods are not always described clearly, which makes further evaluation difficult. The authors do a valiant effort to explain the ideas using schematics, which is helpful. But more clarification is needed. The authors should consider some the following points.

Major points:

1. It is still unclear to me how the authors can definitively point to the presence of a *cis*- versus a *trans*- Notch activation in any particular cell within the early pancreatic bud. They need to make this crystal clear. In addition, they don't consider possible dynamics of the Notch receptors, only using Hes as a proxy. This seems an important omission.

The notion of Dll1 acting via *trans*-activation in the early pancreatic bud is based on observations originally published in Ahnfelt-Rønne et al. (2012) Development 139: 33-45. There we showed that that early buds were reduced in size in *Dll1* mutants and that BrdU incorporation was reduced by ~30% in both E10.5 *Dll1*- and *Hes1* mutant dorsal bud cells. Our interpretation of these results is that Dll1-mediated *trans*-activation is important of normal proliferation.

In contrast, E10.5 *Jag1* mutant buds are increased in size (Seymour et al. 2020), suggesting that *Jag1* acts as a “brake” on Dll1-mediated Notch activation. Importantly, there are two different proposed mechanisms for how *Jag1* can inhibit Dll1-mediated signaling: One is *cis*-inhibition (we propose this in Seymour et al. 2020) and the other is competition for *trans*-activation, assuming that *Jag1* is a weaker *trans*-activator than Dll1. The latter mechanism has previously been proposed to account for increased expression of Notch transcriptional targets and decreased *Ngn3* expression in the embryonic pancreas (Golson et al. (2009) MOD 126: 687-99). However, to our knowledge, no published experiment has to date been able to discriminate between these two potential mechanisms.

We have therefore now analyzed early pancreatic bud size in *Foxa2*^{iCre}-induced conditional *Dll1*; *Jag1* double mutants (*Dll1*; *Jag1*^{ΔFoxa2}) compared to wildtype controls as well as *Dll1*^{ΔFoxa2} and *Jag1*^{ΔFoxa2} single mutants, since the two mechanisms have different predictions for the double mutant phenotype given the interpretation of Dll1 as a *trans*-activator holds true. If *Jag1* acts via *cis*-inhibition, we expect the *Dll1*; *Jag1*^{ΔFoxa2} double mutant to have the same phenotype as the *Dll1*^{ΔFoxa2} single mutant or possibly slightly larger bud size if there is residual *trans*-activating activity present in the double mutant buds (e.g. from ligand expressing endothelial cells or the still uncharacterized Dll4 expression in endocrine progenitors revealed in the many single-cell RNA-seq studies published recently). Conversely, if competition for *trans*-activation is the mechanism by which *Jag1* inhibits Notch activity, then we would expect that the *Dll1*; *Jag1*^{ΔFoxa2} double mutant will be more severely reduced in size than the *Dll1*^{ΔFoxa2} single mutant, as the remaining *Jag1*-

mediated transactivation that is expected to be present in the *Dll1*^{ΔF_{oxa2}} single mutant, albeit weaker than the Dll1-mediated ditto, will be lost in the *Dll1*; *Jag1*^{ΔF_{oxa2}} double mutant.

The outcome of our analysis is quite clear: We observe a slightly increased bud size in *Dll1*; *Jag1* double mutants compared to *Dll1*^{ΔF_{oxa2}} single mutants (new Figure 2a, Response Figure 5), is consistent with the *cis*-inhibition mechanisms but argues against the competition for trans-activation mechanism.

Response Figure 5. Dorsal (left) and ventral (right) pancreas bud volumes relative to control in E10.5 *Dll1*^{ΔF_{oxa2}} and *Jag1*^{ΔF_{oxa2}} single mutants and in *Dll1*; *Jag1*^{ΔF_{oxa2}} double mutants.

Indeed, previous model has included an additional step of modeling the dynamics of the active form of the Notch receptor (NICD), whereas we simplify this step out by approximating that Hes1 is upregulated by the receptor-ligand complex. This simplification is possible as the NICD has a much faster turnover rate (Christy J Fryer, et al. 2004; Neetu Gupta-Rossi, et al. 2001; Camilla Öberg, et al. 2001; Guangyu Wu, et al. 2001; see Methods), allowing us to use a widely accepted method of time-scale separation, where the differential equation for NICD can be set to steady state level and one can use algebraic form instead.

2. Quantification of expression overlaps between Hes, Dll, Jag, Ptf1a, Nkx6.1 is needed from E10.5-12.5.

We have now quantified expression overlaps between Ptf1a and Nkx6-1, Dll1, Jag1 and Hes1 at e10.5 (revised Figure 1b). Since ~95% of the Ptf1a⁺ cells co-express Nkx6-1 at this stage, we chose to quantify the co-expression between Ptf1a and Nkx6-1, Dll1, Jag1 and Hes1 at this stage. For E12.5 where <5% of the Ptf1a cells co-express we quantified the co-expression between Ptf1a and Dll1, Jag1 and Hes1 as well as the co-expression between Nkx6-1 and Dll1, Jag1 and Hes1. As the latter staining included Dll1 and Jag1 in the same staining, we were able to distinguish all

combinations of Dll1 and Jag1 co-expression in the two compartments (revised Figure 1d).

3. Better explanation is needed for methods in Fig.1D (page 6, line 100). Is this analysis of individual cells done on immunofluorescent stained sections? How many cells/sections/embryos? How are the embryos staged? There is mention of bioluminescence and immunofluorescence. Methods are very unclear as is.

The data in the old Figure 1d (revised Figure 1h) are derived from bioluminescence imaging (BLI) of E10.5 pancreatic explants. BLI was performed as For the E10.5 + 1 day experiment n=40 cells from N=3 explants is shown. Similarly, n=83 cells from N=4 explants and n=37 cells from N=3 explants are shown for E10.5 + 4 days and E10.5 + 6 days, respectively. Embryo staging and the BLI procedure including image processing is now explained in the Methods section.

The revised Figure 1h panel show the dynamics of Hes1 protein level in a single cell. Each cell's BLI track was processed this way in order to align the cells in the heatmaps shown in the revised Figure 1g.

Smaller points:

1. No need for "Experimental.." in the figure title for Fig.1. Just "Expression.." would be better.

We thank the reviewer for the comment and have now removed the words "Experimental spatial". Now the title is "Expression patterns of protein associated with cell fate segregation in pancreatic development".

2. It is difficult to appreciate some points made in the introduction, where Figure 1A is mentioned. This panel shows Jag1 staining, while the text refers to Dll1.

We apologize for the errors. We have gone through the text carefully and corrected these sentences.

3. It is difficult to fully appreciate the expression of Jag1/Dll1 in Ptf1a or Hes1 expressing cells without a membrane marker.

We agree with the reviewer that the ligand stainings in particular would have been easier to interpret if the stained samples had included a membrane marker. But given that these stainings are already quadruple IF stainings it was not possible to include a fifth channel. However, we would like to point the reviewer to our previously published co-stainings of the ligands with E-cadherin (Figure S1B and S1C in Seymour et al. 2020).

REVIEWERS' COMMENTS

Reviewer #1 (Remarks to the Author):

The revised manuscript by Xu et al is significantly improved compared to the original manuscript. It provides a much clearer explanations on the experiments and model. The additional experiments (particularly Fig. 2a) and additional analysis of the previous experimental results (Fig. 1) nicely support the model assumptions. Moreover, the additional model analysis including the expansion of the 3D model, the comparison to alternative models, and the sensitivity analysis enhance the confidence in the suggested model. The authors have addressed all the comments we have raised in a satisfactory manner. In particular, our main comment regarding the significance of Dll1 in the model is convincingly addressed. Overall, the manuscript presents a plausible model that explains the observed phenomena and the behavior of the Notch ligands mutants. This model is important since it addresses several important aspects including how oscillatory Hes expression can be resolved by lateral inhibition, and how cis and trans interactions of different ligands can regulate a cell fate decision process.

We do have some minor comments that need some further clarifications:

- 1) A comment about figure 5 d-f, 3 cells are presented and their location is marked in figure b,c. First, the arrows in b,c, are really small and hard to distinguish. Second, since all arrows have the same color, it is not clear which panel correspond to which cell. Please use different colors. In addition, we suggest to use the same y-axis limits for the plots in d-f to allow proper comparison.
- 2) The analysis of time delay in Dll1 transcription is interesting. Please clarify if the 3D model with time delay in Dll1 includes Jag1 or not.
- 3) The authors provided an explanation for the complex term [ND] (as well as other complexes). While the approximation is now clear, we think it is not fully accurate in the limit when $N \sim D$. The authors should state that the approximation is not valid at this limit (it can be seen by not taking the $[ND] \ll N, D$ and then assuming tight binding). We do agree that in most cases the assumptions are valid.
- 4) When discussing alternative model with very fast Ptf1 dynamics, the authors refer to Supp Fig 6e-f saying that "all of the MPC cells would differentiate immediately". However Supp Fig. 6f does seem to show many MPCs. Is it the wrong figure?
- 5) The sentence in the abstract " It suggests that Jag1 cis-interaction is more decisive in cell fate segregation than the weaker trans-activation feedback associated with Dll1." is quite vague. Suggest to change it to a more concrete statement.
- 6) The estimation of the number of neighbors in 3D from 2D cuts should be explained in detail in the methods.
- 7) Supp Figs 7 and 8 are discussed for the first time in the discussion. We suggest to move these to the end of the results.

Reviewer #2 (Remarks to the Author):

I strongly appreciate the big effort made by the authors to address the reviewers' concerns. The revised manuscript contains many new data and many new simulations. The authors have provided more confirmation of the assumptions on which the model relies and have clarified the issues raised (both from the experimental data and from the modelling part). I agree with most of the responses provided by the authors. Yet I have the following concerns regarding few of them:

- A) The authors claim that by assessing the phenotype when Dll1 and Jag1 are both deleted, it can be discerned between whether Jag1 inhibits Dll1-mediated signaling through cis-inhibition or through competition for trans-activation. I am not fully convinced with the interpretation. In my opinion, the double mutant can not be used to clearly discern on how Jag1 is inhibiting Dll1-mediated signaling. The authors indicate that if Jag1 acts through cis-inhibition, the double mutant should have the same phenotype as the Dll1 mutant, or a phenotype according to slightly higher notch activation than the

Dll1 mutant, since *there can be residual trans-activation from other ligands. I agree with this. Then, they say that if Jag1 acts by competing with Dll1 for trans-activation, then the double mutant should have a phenotype with less Notch activation than that of Dll1 mutant. The reason is that the signaling of Jag1 that can be acting in the Dll1 mutant will be absent in the double mutant. I also agree with all that. But, as claimed in *, the double mutant can have residual trans-activation from other ligands. If this residual trans-activation is higher than the signaling of Jag1 in the Dll1 mutant, then the double mutant can have a phenotype consistent with higher Notch activation than the Dll1 mutant (and so it can have the same phenotype expected from Jag1 acting through cis-inhibition). Why the residual trans-activation can not be higher than the signaling of Jag1 in the Dll1 mutant? In the competition for trans-activation mechanism, Jag1 is expected/assumed to have a weak signaling, which may be less than the residual one. Perhaps this is not the most plausible scenario but additional arguments should be provided.

In addition, I think that all these reasonings when considering that Jag1 inhibits through cis-inhibition assume a higher affinity of Jag1 for cis-binding than that of Dll1. However, in the model, for simplicity, the same affinity of cis-binding is assumed for Dll1 and Jag1. It may be worth clarifying this.

B) The authors indicate that because NICD has a fast turnover, then separation of time scales can be used and then NICD dynamics are not explicitly modelled (instead, activation into Hes1 dynamics is settled). However, NICD has a mean lifetime of 90-180min (Sprinzak and Blacklow , Annu Review Biophys 2021). In ref. 52 it is indicated that NICD can have a half-life of 45 min when MAM is present. Since the degradation rate of Dll1 is estimated to be 50 min^{-1} (Table from manuscript), I do not see that the approximation of time scale separation is justified. The results may not depend on whether the assumption is made or not, but better justification is required.

The manuscript should be revised for missing end points and capital letters. In addition:

Line 105: "suggest" should be changed to suggests

Fig.2 f: the dynamics of Hes1 in Cell2 are "cut" at early times. Better to use a larger range in vertical axis.

Fig.5c caption: please indicate that the dynamics of three cells (and not of two interacting cells) are shown.

point-by-point response to reviewers' comments

REVIEWERS' COMMENTS

Reviewer #1 (Remarks to the Author):

The revised manuscript by Xu et al is significantly improved compared to the original manuscript. It provides a much clearer explanations on the experiments and model. The additional experiments (particularly Fig. 2a) and additional analysis of the previous experimental results (Fig. 1) nicely support the model assumptions. Moreover, the additional model analysis including the expansion of the 3D model, the comparison to alternative models, and the sensitivity analysis enhance the confidence in the suggested model. The authors have addressed all the comments we have raised in a satisfactory manner. In particular, our main comment regarding the significance of Dll1 in the model is convincingly addressed. Overall, the manuscript presents a plausible model that explains the observed phenomena and the behavior of the Notch ligands mutants. This model is important since it addresses several important aspects including how oscillatory Hes expression can be resolved by lateral inhibition, and how cis and trans interactions of different ligands can regulate a cell fate decision process.

We do have some minor comments that need some further clarifications:

1) A comment about figure 5 d-f, 3 cells are presented and their location is marked in figure b,c. First, the arrows in b,c, are really small and hard to distinguish. Second, since all arrows have the same color, it is not clear which panel correspond to which cell. Please use different colors. In addition, we suggest to use the same y-axis limits for the plots in d-f to allow proper comparison.

We thank the reviewer for the suggestion. We change the color of the arrows in the figures (Fig. 5b, c and Fig. 3b) and also put the arrows along with the corresponding gene expression dynamics (Fig. 5 d–f and Fig. 3c). Now the arrows are very easy to distinguish and make the figures clear. The range of y-axis in Fig. 5 d–f is consistent now if the cells show oscillations.

2) The analysis of time delay in Dll1 transcription is interesting. Please clarify if the 3D model with time delay in Dll1 includes Jag1 or not.

We thank the reviewer for reminding us. The 3D model is done with Jag1 and mimics the scenario in pancreatic development. To clary it, we now add a sentence in the figure caption (Supplementary Fig. 4) "Simulations in e–i are done with Jag1 in the model". We also modified the manuscript where discuss about it to make it clear.

3) The authors provided an explanation for the complex term [ND] (as well as other complexes). While the approximation is now clear, we think it is not fully accurate in the limit when $N \sim D$. The authors should state that the approximation is not valid at this limit (it can be seen by not taking the $[ND] \ll N, D$ and then assuming tight binding). We do agree that in most cases the assumptions are valid.

We thank the reviewer for pointing it out. We now state the limit in the Methods that: The simplified calculation of [ND] is valid for the case of pancreatic cells where $D \ll N$. This

simplification may not be applicable for the systems where the concentrations of receptors and ligands are comparable.

4) When discussing alternative model with very fast Ptf1 dynamics, the authors refer to Supp Fig 6e-f saying that “all of the MPC cells would differentiate immediately”. However Supp Fig. 6f does seem to show many MPCs. Is it the wrong figure?

We apologize for this misleading sentence. In the revised manuscript, we correct it. The figure is correct. We modified this sentence to: “The time window before the differentiation of MPCs becomes very short (Supplementary Fig. 6e–6f), which indicates no pancreas development since the tissue can not have proper MPC expansion.”

5) The sentence in the abstract “ It suggests that Jag1 cis-interaction is more decisive in cell fate segregation than the weaker trans-activation feedback associated with Dll1.” is quite vague. Suggest to change it to a more concrete statement.

We thank the reviewer for the suggestion. We rewrite this sentence to: “It suggests that *cis*-interaction is crucial for exiting the multipotent state, while *trans*-interaction is required for adopting the bipotent fate.”

6) The estimation of the number of neighbors in 3D from 2D cuts should be explained in detail in the methods.

We thank the reviewer for the suggestion. In the revised Methods section, we explained how we estimated the number of neighbors in 3D.

7) Supp Figs 7 and 8 are discussed for the first time in the discussion. We suggest to move these to the end of the results.

We agree with the reviewer and organized these results to the end of the Results section.

Reviewer #2 (Remarks to the Author):

I strongly appreciate the big effort made by the authors to address the reviewers' concerns. The revised manuscript contains many new data and many new simulations. The authors have provided more confirmation of the assumptions on which the model relies and have clarified the issues raised (both from the experimental data and from the modelling part). I agree with most of the responses provided by the authors. Yet I have the following concerns regarding few of them:

A) The authors claim that by assessing the phenotype when Dll1 and Jag1 are both deleted, it can be discerned between whether Jag1 inhibits Dll1-mediated signaling through cis-inhibition or through competition for trans-activation. I am not fully convinced with the interpretation. In my opinion, the double mutant can not be used to clearly discern on how Jag1 is inhibiting Dll1-mediated signaling. The authors indicate that if

Jag1 acts through cis-inhibition, the double mutant should have the same phenotype as the Dll1 mutant, or a phenotype according to slightly higher notch activation than the Dll1 mutant, since *there can be residual trans-activation from other ligands. I agree with this. Then, they say that if Jag1 acts by competing with Dll1 for trans-activation, then the double mutant should have a phenotype with less Notch activation than that of Dll1 mutant. The reason is that the signaling of Jag1 that can be acting in the Dll1 mutant will be absent in the double mutant. I also agree with all that. But, as claimed in *, the double mutant can have residual trans-activation from other ligands. If this residual trans-activation is higher than the signaling of Jag1 in the Dll1 mutant, then the double mutant can have a phenotype consistent with higher Notch activation than the Dll1 mutant (and so it can have the same phenotype expected from Jag1 acting through cis-inhibition). Why the residual trans-activation can not be higher than the signaling of Jag1 in the Dll1 mutant? In the competition for trans-activation mechanism, Jag1 is expected/assumed to have a weak signaling, which may be less than the residual one. Perhaps this is not the most plausible scenario but additional arguments should be provided.

We are pleased that the reviewer agrees with our reasoning in general, but also agree that additional arguments would strengthen our reasoning in the case where residual transactivation from other ligands may influence the outcome. These additional arguments are as follows: First, we would like to clarify that there is only one ligand that can contribute with residual transactivation, namely Dll4, as Jag2 expression has never been detected in the fetal pancreas. However, RNA data suggest that within the pancreas epithelium *Dll4* is only expressed in the relatively scarce Ngn3+ endocrine precursors (Larsen et al. 2017 Nat Commun 8:605; Scavuzzo et al. 2018 *ibid* 9:3356; Byrnes et al. 2018 *ibid* 9: 3922), and we find the same distribution for the Dll4 protein by IF analysis (Figure_A, unpublished).

E10.5

Figure_A. Notch ligand expression in E10.5 dorsal pancreas. Note that within the pancreas epithelium, Dll4 is expressed only in relatively few Ngn3+ endocrine precursors (arrows), compared to the broad expression of Dll1 in both Ngn3+ endocrine precursors, Gcg+ endocrine cells and Ptf1a+Sox9+ MPCs. As expected Dll4 is also found in Pecam+ endothelial cells (ec).

Due to the relatively low number of Dll4+Ngn3+ cells compared to Dll1+Sox9+Ptf1a+ MPCs, we assume that Dll4 only makes a minor contribution to the Notch transactivation in the E10.5 MPCs. This assumption is supported by preliminary data from Pdx1-Cre-mediated conditional Dll1; Dll4 single- and double knockouts that show no significant effect of Dll4 deficiency on E10.5 dorsal bud volume, even when Dll1 is deficient (Figure_B, unpublished).

dp volume E10.5: Pdx1-Cre x Dll1Dll4 floxed

Figure_B. Quantification of E10.5 dorsal bud volumes in the indicated genotypes, shown as % of controls. Controls (CTR) are Cre-negative littermates. Note that $Dll1^{+/+}; Dll4^{f/f}$ ($Dll4^{f/f}$) is not different from controls and that $Dll1^{f/f}; Dll4^{f/f}$ is not different from $Dll1^{f/f}; Dll4^{f/+}$. We are currently collecting $Dll1^{f/f}; Dll4^{+/+}$ embryos to complete this analysis. However, note that the dorsal bud size reduction in $Pdx1-Cre; Dll1^{f/f}; Dll4^{f/+}$ embryos is comparable to the reduction seen in $Foxa2-iCre; Dll1^{f/f}$ embryos (Fig. 2A in main manuscript). Adjusted *p*-values were calculated in Graphpad Prism by a One-way ANOVA test with Tukey's post-hoc analysis for multiple comparisons.

We now discuss these additional arguments in the revised manuscript and furthermore have modified our conclusion to state that: "The most parsimonious explanation is that Jag1 act by *cis*-inhibition.

In addition, I think that all these reasonings when considering that Jag1 inhibits through *cis*-inhibition assume a higher affinity of Jag1 for *cis*-binding than that of Dll1.

However, in the model, for simplicity, the same affinity of *cis*-binding is assumed for Dll1 and Jag1. It may be worth clarifying this.

We agree with the reviewer on this point also and have added a paragraph to the discussion section in the revised manuscript clarifying this.

B) The authors indicate that because NICD has a fast turnover, then separation of time scales can be used and then NICD dynamics are not explicitly modelled (instead, activation into Hes1 dynamics is settled).

However, NICD has a mean lifetime of 90-180min (Sprinzak and Blacklow , Annu Review Biophys 2021). In ref. 52 it is indicated that NICD can have a half-life of 45 min when MAM is present.

Since the degradation rate of Dll1 is estimated to be 50 min^{-1} (Table from manuscript), I do not see that the approximation of time scale separation is justified.

The results may not depend on whether the assumption is made or not, but better justification is required.

We thank the reviewer for pointing out better justification is required for this part in our model. However, we believe that the NICD half-life values measured for ectopically expressed NICD are not as straightforward to interpret as they might appear. The $t_{1/2}$ of ectopically expressed NICD seems to vary greatly between cell lines and even within the same cell line in different experiments (Ilagan et al. 2011, *Science Signaling* 4, rs7; Kuang et al. 2020, eLife 9:e53659). Furthermore, as mentioned by the referee, previous work shows that co-expression of MAM reduced NICD $t_{1/2}$ from 180 min. to 45 min. (Fryer et al. 2004, Mol Cell 16:509), which raises concerns about whether other components may be rate limiting in the situation where NICD (and MAM) are arguably overexpressed.

Moreover, Kuang et al. 2020 demonstrates that NICD is degraded very fast once it sits on a target gene promoter – the Cdk8-dependent “bind and discard” mechanism. Therefore, one can ask whether these measured half-lives of ectopically expressed NICD are the best input parameters for our model. We would like to argue that it is more appropriate to consider the $t_{1/2}$ of endogenous NICD, co-precipitated with Su(H). This was reported by the Bray group to be ~ 10 min. (Housden et al. 2013, PLOS Genetics 9:e1003162).

We have therefore now compared our simplified model with a model including an additional variable, $t_{1/2}$ NICD (see Figure C below). The two models have the same cell fate differentiation results, just exactly as the reviewer’s expected. But there is some difference when NICD decay is slow ($\tau_I = 45$ min). The oscillation phases between the two cells shifts from anti-phase to become in-phase when NICD $t_{1/2}$ is >45 min. In contrast, a $\tau_I = 10$ min. gives the same result as the simplified model.

We therefore modified this part in the Methods to provide a better justification that includes the above-mentioned considerations on NICD half-life measurements.

Fig. C. A model including NICD $t_{1/2}$. The dynamic of NICD in the model is described by an independent equation (dl/dt) with two extra parameters: production rate (a_I) and degradation time (τ_I) in minutes. With the two-cell model, the simulation shows similar dynamics to the simplified one in the manuscript with the fast degradation time scale of NICD ($\tau_I = 1$ min and $\tau_I = 10$ min). With the slow degradation ($\tau_I = 45$ min), the cell differentiation does not change at the end, while the anti-phase oscillations at the transient MPC state become in-phase. In these three simulations, a_I is fitted, other parameters (except τ_I) are the same as used in the manuscript.

The manuscript should be revised for missing end points and capital letters. We thank the reviewer for reminding of these typos. We carefully went through the manuscript and corrected them.

In addition:
Line 105: “suggest” should be changed to suggests

Thanks, we corrected it in the revised manuscript.

Fig.2 f: the dynamics of Hes1 in Cell2 are “cut” at early times. Better to use a larger range in vertical axis.

We thank the reviewer for pointing this out. We modified the figure and enlarged the range of y-axis. Now the full curve is shown.

Fig.5c caption: please indicate that the dynamics of three cells (and not of two interacting cells) are shown.

Thanks for the reviewer, we correct it in the caption.